

# Leveraging a time-series event separation method to untangle time-varying hydrologic controls influence on wildfire disturbed streamflow

Haley A. Canham[1], Belize A. Lane[1], Colin B. Phillips[1], Brendan P. Murphy[2]

[1] Department of Civil and Environmental Engineering, Utah State University, Logan, UT, US
    [2] School of Environmental Science, Simon Fraser University, Burnaby, BC, Canada

*Correspondence to:* Haley A. Canham (haley.canham@usu.edu)

**Abstract.** Watershed disturbances can have broad, long-lasting impacts that result in a range of streamflow response. Increasing disturbance regimes, particularly from wildfire, is a growing concern for watershed management. The
influence of watershed disturbances on rainfall-runoff patterns has proved challenging to isolate from undisturbed streamflow variability due to the role of hydrologic controls that vary through time, including water year type, seasonality, and antecedent precipitation. To better assess the influence of watershed disturbance on rainfall-runoff event patterns we developed the Rainfall-Runoff Event Detection and Identification (RREDI) toolkit. The RREDI toolkit is a novel time-series event separation method that automates the pairing and attribution of precipitation and
streamflow events, leveraging and building on existing event separation methods. A rainfall-runoff event dataset of 5042 events was generated by the RREDI toolkit from a collection of nine western US study watersheds spanning a range of streamflow regimes, watershed properties, and burn characteristics. Through analyzing the rainfall-runoff event dataset, we found that water year type and season were significant controls on rainfall-runoff metrics. The significance of antecedent precipitation was variable between watersheds, indicating a more complex relationship for
this control. The watershed-specific permutations of significant controls resulted in unique significant condition group trends in the rainfall storm depth and peak runoff relationship in two contrasting watersheds. In general, for each of the significant condition groups post-fire peak runoff was higher than undisturbed peak runoff except during winter in snow-dominated watersheds. Consideration of the time-varying hydrologic controls, particularly water year type and season, were identified as important when untangling the influence of wildfire on the rainfall-runoff patterns. The
RREDI toolkit can be further applied to investigate the influence of other watershed disturbances and controls to increase understanding of rainfall-runoff patterns across the landscape.

## 1. Introduction

Watershed disturbances can have broad, long lasting, and variable impacts on watershed hydrology (Ebel & Mirus, 2014). A range of disturbances including wildfire, drought, flood, insect infestation, invasive species,
agriculture,  urbanization, mining, and forest management have been observed to alter streamflow (Adams et al., 2012; Brantley et al., 2013; Ebel & Mirus, 2014; Goeking & Tarboton, 2020; Hopkins et al., 2015; Kelly et al., 2017; Miller & Zégre, 2016). Wildfire is particularly impactful: since 2000 an average of 7.0 million acres has burned annually in the US (Hoover & Hanson, 2021). Further, in a changing climate the occurrence and severity of wildfire is increasing (Hallema et al., 2018; Murphy et al., 2018; Oakley, 2021; Robinne et al., 2021). Distilling the influence of watershed



disturbance from the natural variability within streamflow has proved challenging across disturbance regimes. A better understanding of hydrologic controls that vary in time in disturbed watersheds is critical for watershed management resiliency in the face of increasing disturbance regimes (Mirus et al., 2017).

Wildfires can cause abrupt changes to hydrologic processes and properties resulting in altered streamflow patterns that change through time as the watershed recovers (Ebel & Mirus, 2014; Santi & Rengers, 2020; Wagenbrenner et

al., 2021). Post-fire changes in soil properties and vegetation may alter runoff generation processes (Ebel, Moody, et al., 2012; Santi & Rengers, 2020). Altered soil properties may include changes in soil-water repellency and infiltration capacity, the presence of ash, and loss of soil organic matter (Balfour et al., 2014; Ebel, Moody, et al., 2012; Santi & Rengers, 2020). Loss of vegetation may alter evapotranspiration and interception within the watershed (Atchley et al., 2018; Poon & Kinoshita, 2018; Santi & Rengers, 2020). The observed influence of these altered hydrologic properties

on streamflow is variable in both the direction and magnitude of change. Total annual streamflow has been observed to increase (Bart, 2016; Beyene et al., 2021; Caldwell et al., 2020; Y. Guo et al., 2021; Hallema et al., 2017; Khaledi et al., 2022; Kinoshita & Hogue, 2015; Mahat et al., 2016; Owens et al., 2013; Saxe et al., 2018; Wine et al., 2018; Wine & Cadol, 2016), decrease (Balocchi et al., 2020; Biederman et al., 2022), and stay the same (Bart & Hope, 2010; Vore et al., 2020). Post-fire event flows have similarly been found to increase (Beyene et al., 2021; Hallema et al.,

2017; Mahat et al., 2016; Saxe et al., 2018), decrease (Balocchi et al., 2020), and show no significant change (Kinoshita & Hogue, 2015; Long & Chang, 2022; Newcomer et al., 2023; Nunes et al., 2020; Owens et al., 2013). This leaves questions about our ability to distill the influence of the wildfire disturbance from the watershed natural variability.

In addition to watershed disturbances, time-varying hydrologic controls including water year type (WYT), seasonality, and antecedent precipitation have been found to influence rainfall-runoff patterns. Water year type is a

commonly used categorization to compare individual years against historical trends (Null & Viers, 2013). Variation between WYT wet and dry years may result in differences in runoff response (Biederman et al., 2022; Null & Viers, 2013). Examples of WYT variation drivers include variation in annual snowpack (Cayan, 1995) or the occurrence and intensity of precipitation from monsoons or atmospheric rivers (Arriaga-Ramierez & Cavazos, 2010; Pascolini-Campbell et al., 2015). Seasonality, specifically seasons based on the annual hydrograph, can alter event runoff

response across a range of watersheds (Merz et al., 2006; Merz & Blöschl, 2009; Norbiato et al., 2009; Tarasova, Basso, Zink, et al., 2018). Seasonal differences have been attributed to precipitation type, storm properties (intensity, depth), water balance, and antecedent wetness conditions (Berghuijs et al., 2014; Merz et al., 2006; Merz & Blöschl, 2009; Norbiato et al., 2009; Tarasova, Basso, Zink, et al., 2018). Finally, antecedent precipitation and antecedent moisture have been found to alter event runoff response (Merz et al., 2006; Merz & Blöschl, 2009; Tarasova, Basso,

Zink, et al., 2018). Antecedent precipitation is commonly used as a proxy for antecedent moisture (Merz & Blöschl, 2009; Mishra & Singh, 2003; Tarasova, Basso, Zink, et al., 2018). Despite their established influence on event runoff response, these time-varying hydrologic controls are inconsistently considered in hydrologic disturbance studies.

Selected post-fire streamflow change studies have assessed some of these time-varying hydrologic controls, but to the best of the authors knowledge none to date have considered all three potential controls and very few studies

have focused on the event scale. Of these three controls, WYT is most frequently considered when evaluating wildfire influence on streamflow (Beyene et al., 2021; Biederman et al., 2022; Hallema et al., 2017; Long & Chang, 2022;



Wine & Cadol, 2016). A common method to account for the role of WYT variability is compare water year expected streamflow and observed streamflow to isolate the influence of the disturbance (Beyene et al., 2021; D. Guo et al., 2023; Hallema et al., 2017; Mahat et al., 2016; Newcomer et al., 2023). Another method for pre- and post-fire

comparison is water year typing based on total annual precipitation-streamflow relationships or annual percentiles (Biederman et al., 2022; Long & Chang, 2022). In addition to interannual variability, several studies have evaluated post-fire changes in total streamflow or flow statistics within specific seasons (Balocchi et al., 2020; Biederman et al., 2022; Kinoshita & Hogue, 2015; Saxe et al., 2018; Wine et al., 2018). Antecedent precipitation is less commonly considered in post-fire streamflow response studies. Long & Chang (2022) used three-day antecedent precipitation to

normalized runoff event volume, but found no altered streamflow significance. Lack of consistent consideration of WYT, seasonal variability, and antecedent precipitation may help explain the inconsistency in observed post-fire effects on streamflow.

Large-sample hydrology studies are frequently used to investigate time-varying and static watershed controls on event-scale rainfall-runoff patterns. The event-scale enables a process-based understanding of driving hydrologic

processes in catchment hydrology (Gupta et al., 2014; Sivapalan, 2009). Large-sample investigations into event-scale controls in Europe have found that time-varying hydrologic controls influence event runoff ratios (Merz et al., 2006; Merz & Blöschl, 2009; Norbiato et al., 2009; Tarasova, Basso, Poncelet, et al., 2018; Tarasova, Basso, Zink, et al., 2018). A similar event-scale large-sample study of 432 US watersheds evaluated only static controls on event runoff response, and identified aridity, topographic slope, soil permeability, rock type, and vegetation density as significant

(Wu et al., 2021). None of these studies considered the separate impact of watershed disturbance. Long & Chang (2022) considered WYT and antecedent precipitation while investigating the influence of wildfire disturbance on event runoff response. However, they analyzed only a small-sample of rainfall-runoff events from two years, one pre- and one post-fire, in a small-sample of six watersheds in Oregon, US.

The objectives of this paper were twofold. The first was to describe and evaluate the Rainfall-Runoff Event

Detection and Identification (RREDI) toolkit, a novel time-series event separation method (Canham & Lane, 2022). The second was to apply the proposed method to investigate the influence of time-varying hydrologic controls including WYT, season, and antecedent precipitation on event runoff response. The specific aims of the investigation into time-varying hydrologic controls were to (1) explore rainfall-runoff patterns, (2) identify significant time-varying hydrologic controls on event runoff response, and (3) evaluate how time-varying hydrologic controls influence event

runoff response in wildfire disturbed watersheds. We hypothesize that accounting for these time-varying hydrologic controls will untangle the natural watershed streamflow variability thereby making the influence of the disturbance more apparent.

## 2. Study watersheds

Nine study watersheds in the western US were selected based on watershed properties, burn characteristics, and

streamflow data availability. First, we identified western US watersheds with at least 20 years of continuous 15-minute streamflow data (Falcone, 2011). Of these, we identified watersheds with a greater than 5% area burned within the available record of 1984 to 2020 from the MTBS database (*Monitoring Trends in Burn Severity (MTBS)*, n.d.). This





set was further reduced to watersheds with pre-fire and post-fire streamflow records of at least ten and six years respectively and minimal upstream anthropogenic influence, such as reservoirs. The watersheds spanned five magnitudes of contributing area (1.8 to 10,125 km$^2$) and from 5 to 100% area burned. The final study watershed selection from this set were those with no other fires within the MTBS database exceeding 5% area burned within the watershed. The nine selected watersheds spanned a wide range of watershed properties and burn characteristics (Figure 1).

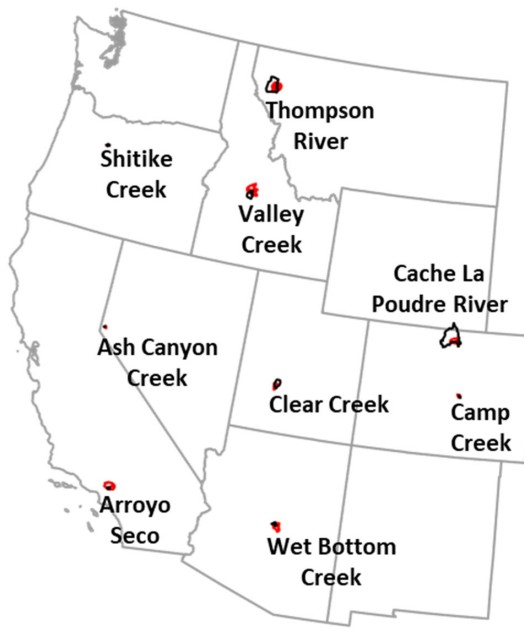


**Figure 1: Nine selected study watersheds (labeled). Shown are watersheds (black) and fire perimeters (red).**

The nine study watersheds spanned a large range of contributing areas, streamflow regimes, and burn conditions (Table 1). The contributing area range was three orders of magnitude, where the largest watershed, the Cache La Poudre River, was 2,966 km$^2$, and the smallest, Ash Canyon Creek, was 14 km$^2$. The watersheds included a range of streamflow regimes including seven snow melt dominated systems with average annual hydrograph peak dates between April and June and two wet season rain dominated systems with average annual hydrograph peak dates between January and February. All nine study watersheds were impacted with differing size and severity fires. The highest impacted was Arroyo Seco from the Station Fire (2009) with 100% area burned (78% high and moderate burn severity). The least impacted watershed was the Cache La Poudre River from the High Park Fire (2012) with 10% area burned (5% high and moderate severity).



**Table 1: Watershed properties and burn characteristics for the study watersheds.**

| Watershed | State | Contributing area (km$^2$) | USGS Gage ID | Fire | Ignition Year | Area burned (%) |
|---|---|---|---|---|---|---|
| Arroyo Seco | CA | 42 | 11098000 | Station Fire | 2009 | 100 |
| Ash Canyon Creek | NV | 14 | 10311200 | Waterfall Fire | 2004 | 63 |
| Cache La Poudre | CO | 2966 | 06752260 | High Park Fire | 2012 | 10 |
| Camp Creek | CO | 25 | 07103703 | Waldo Canyon Fire | 2012 | 85 |
| Clear Creek | UT | 426 | 10194200 | Twitchell Canyon Fire | 2010 | 25 |
| Shitike Creek | OR | 57 | 14092750 | Waterfalls 2 Fire | 2012 | 76 |
| Thompson River | MT | 1652 | 12389500 | Chippy Creek Fire | 2007 | 15 |
| Valley Creek | ID | 376 | 13295000 | Halstead Fire | 2012 | 27 |
| Wet Bottom Creek | AZ | 94 | 09508300 | Willow Fire | 2004 | 84 |

### 2.1. Hydrologic data inputs

Streamflow and precipitation data were obtained for each study watershed. The 15-minute, daily, and total annual streamflow data for the full period of record were retrieved from the USGS streamflow gage. The total annual precipitation at the centroid of each study watershed used to classify WYT was retrieved from PRISM (Oregon State University, 2022). Hourly precipitation time series were obtained for the watershed centroid from the Analysis of Record Calibration (AORC) 4km$^2$ resolution data product for water years 1980 to 2022 (National Weather Service

Office of Water Prediction, 2021). Linear interpolation was used to spread the hourly rainfall over the timestep at the AORC resolution of 1 mm. The AORC data product was selected because of higher correlation between the AORC data product and rain gage measurements compared to other gridded precipitation data products in studies in Louisiana and the Great Lakes basins (Hong et al., 2022; Kim & Villarini, 2022). We additionally performed a comparison of storm events in a mountainous region, specifically in Clear Creek watershed, for water year 2011. We compared the

AORC-based storm events at the corresponding locations of two rain gages: one temporary rain gage in the mountains installed after a wildfire (Murphy et al., 2019) and a NOAA COOP rain gage in the nearby valley 18 km from Clear Creek watershed (Beaver 4E, UT US COOP:420522). For the storm events in water year 2011, the average storm depth based on the AORC data product was less than that measured at the rain gage by 0.6 mm for the Clear Creek rain gage and by 8.3 mm for the NOAA COOP rain gage. Similarly, the average 60-minute peak storm intensity for

the AORC data product was less than the rain gage by 2.3 mm/hr for the Clear Creek rain gage and by 1.5 mm/hr for the NOAA COOP rain gage.

### 3. Methods

We describe the four key steps of the RREDI toolkit in section 3.1 (Figure 2), with additional details in Supplemental Information. A rainfall-runoff event dataset was created by applying the RREDI toolkit to nine western





US watersheds (Figure 2). This dataset was then used to explore rainfall-runoff event patterns, identify significant time-varying hydrologic controls, and evaluate the influence of these controls in wildfire disturbed watersheds (Figure 2). The hydrologic conditions associated with each time-varying hydrologic control were identified and assigned for each event as described in section 3.2. The sorted rainfall-runoff events were then explored as described in section 3.3. Trends in rainfall-runoff event patterns were identified using a LOWESS curve. Inferential statistics were used to test the significance of the hydrologic conditions to identify the significant time-varying hydrologic controls for generalized runoff metric groups. The influence of the wildfire was then evaluated relative to the undisturbed runoff event significant condition group trends in two contrasting watersheds, Arroyo Seco and Clear Creek.

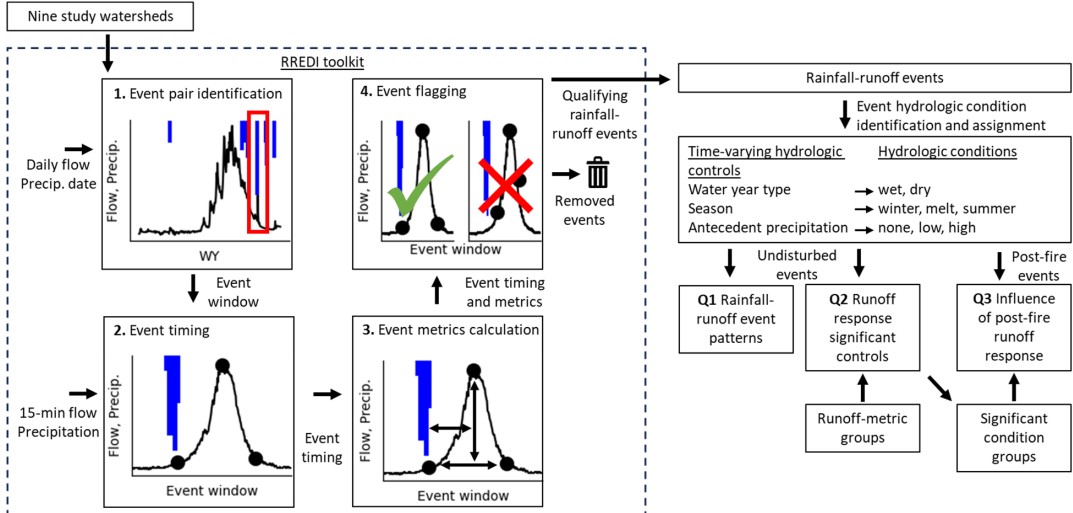

**Figure 2: Methods workflow to explore the influence of time-varying hydrologic controls on rainfall-runoff event patterns as described in this paper. The four key steps of the RREDI toolkit (black dashed box) are outlined: Step 1. Event pair identification, Step 2. Event timing, Step 3. Event metrics calculation, and Step 4. Event flagging. Major connections between workflow steps and study research questions (Q) are shown.**

### 3.1. RREDI toolkit

The RREDI toolkit was developed to automatically separate rainfall-runoff events for any watershed using time-series signal processing in four steps (Figure 2; Supplemental Information) (Canham & Lane, 2022). Signal processing theory provided techniques including data smoothing, peak detection, and window processing that were used to automate detection of features from a time series (Patterson et al., 2020). In step 1, rainfall-runoff event pairs and the associated event window were identified using daily streamflow and precipitation data based on the co-occurrence of separately identified rainfall events by separating precipitation timeseries into storms and runoff events using signal processing theory from the overlapping period of record (Figure 2). For each event pair, the event window from the start of the rainfall to the end of runoff was passed to step 2. The runoff event start, peak, and end timing and magnitude



and the runoff event volume were then identified using then 15-minute streamflow data and the 60-minute storm intensity in step 2 (Figure 2; Figure 3). For each rainfall-runoff event, a set of 17 metrics were calculated using the rainfall and runoff event timings in step 3 (Figure 2). Metrics fell within four runoff metric groups: runoff volume metrics, runoff magnitude metrics, runoff duration metrics, and rainfall-runoff timing metrics (Fig. S4; Table S3). Selected metrics utilized further in this study included those as follows (Figure 3 b). The runoff volume metric group included event volume. The runoff magnitude metric group included runoff peak defined by the runoff peak magnitude. The runoff duration metric group included event duration calculated as the difference between the runoff event start and end times. The rainfall-runoff timing metric group included response time calculated as the difference between the storm start time and the runoff start time. Metrics were also normalized by contributing area to facilitate comparison between study watersheds. Finally, in step 4, event flagging was performed to remove incorrectly identified events falling within four event identification issues: gaps in 15-minute streamflow data, diurnal cycling, duplicate events, and no identified event end time (Figure 2; Supplemental Information). The RREDI toolkit was fully automated using the open-source python language.

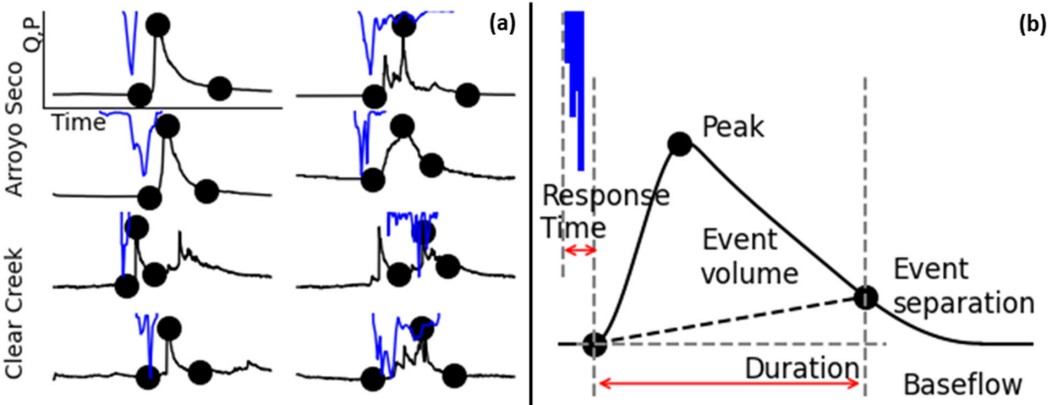

**Figure 3. RREDI toolkit rainfall-runoff event examples and metrics. (a) Eight example rainfall-runoff events identified using the RREDI toolkit. Shown are the rainfall event (blue), the paired runoff event hydrograph (black), and the identified runoff start, peak, and end times and magnitudes (black dots). (b) Example event showing relevant rainfall-runoff event metrics including event volume, peak, duration, and response time. Event separation (black dashed) between event flow volume and baseflow is shown.**

A systematic assessment of the RREDI toolkit outputs was performed. For each of the study watersheds, all rainfall-runoff events occurring in the wettest, mean, and driest water years based on watershed average annual total precipitation (Oregon State University, 2022) were visually inspected. The RREDI toolkit identified runoff start, peak, and end timing were compared with those identified by visual inspection and assessed with respect to the four event identification issues described above. RREDI toolkit performance assessment results for each watershed and overall



included the percent of events with accurately identified timing output from the RREDI toolkit, the percent of events
        flagged in step 4, and the percent of events retained after removal of flagged events.

### 3.2. Hydrologic condition identification and assignment

        Hydrologic conditions were identified and assigned for each rainfall-runoff event with respect to the three time-
        varying hydrologic controls considered in this study: WYT, season, and antecedent precipitation. Events were also
defined as undisturbed or disturbed, where disturbed events were those occurring within six years post-fire (Ebel et
        al., 2022; Wagenbrenner et al., 2021). Water year type was assigned as wet or dry based on total annual streamflow
        and watershed average total annual precipitation following Biederman et al. (2022) for all study watersheds (Figure 4
        a; Fig. S6). Total annual streamflow and precipitation were plotted for the undisturbed period of record to visually
        identify the annual precipitation threshold above which streamflow increased linearly with precipitation. Years (both
undisturbed and disturbed) with annual precipitation above or below the threshold were then classified as wet or dry,
        respectively. For watersheds where no precipitation threshold was identified, the driest third of years (both undisturbed
        and disturbed) by annual precipitation were considered dry. Winter, melt, and summer hydrologic seasons were
        identified for each watershed based on inspection of the average annual hydrograph and the earliest and latest mean
        (2001 - 2018) snow-off dates within the watershed (O'Leary III et al., 2020) (Figure 4 b; Fig. S7). The start of winter
season was uniformly set as November 1 to capture the change in precipitation pattern and type between summer and
        winter. Melt season started the month after the earliest snow-off date and summer season started the month after the
        latest snow-off date to account for the lagged streamflow response to snowmelt. Watersheds with less than 10% of the
        watershed with an identified snow melt date were considered to have no melt season (i.e., only winter and summer).
        In watersheds with no melt, summer started the month where baseflow dominated over winter storm peaks in the mean
annual hydrograph. Event antecedent precipitation was assigned as none (<1mm), low (1-25mm), and high (>25mm)
        based on the cumulative precipitation depth over the six days prior to the precipitation event start time (Long & Chang,
        2022; Merz et al., 2006; Merz & Blöschl, 2009; Tarasova, Basso, Zink, et al., 2018) (Figure 4 c). Only snow-off events
        were considered in this assessment, including only summer events in watersheds with a melt season and all events in
        watersheds without a melt season, to isolate the influence of soil moisture on runoff rather than snowmelt and rain-
on-snow influences.

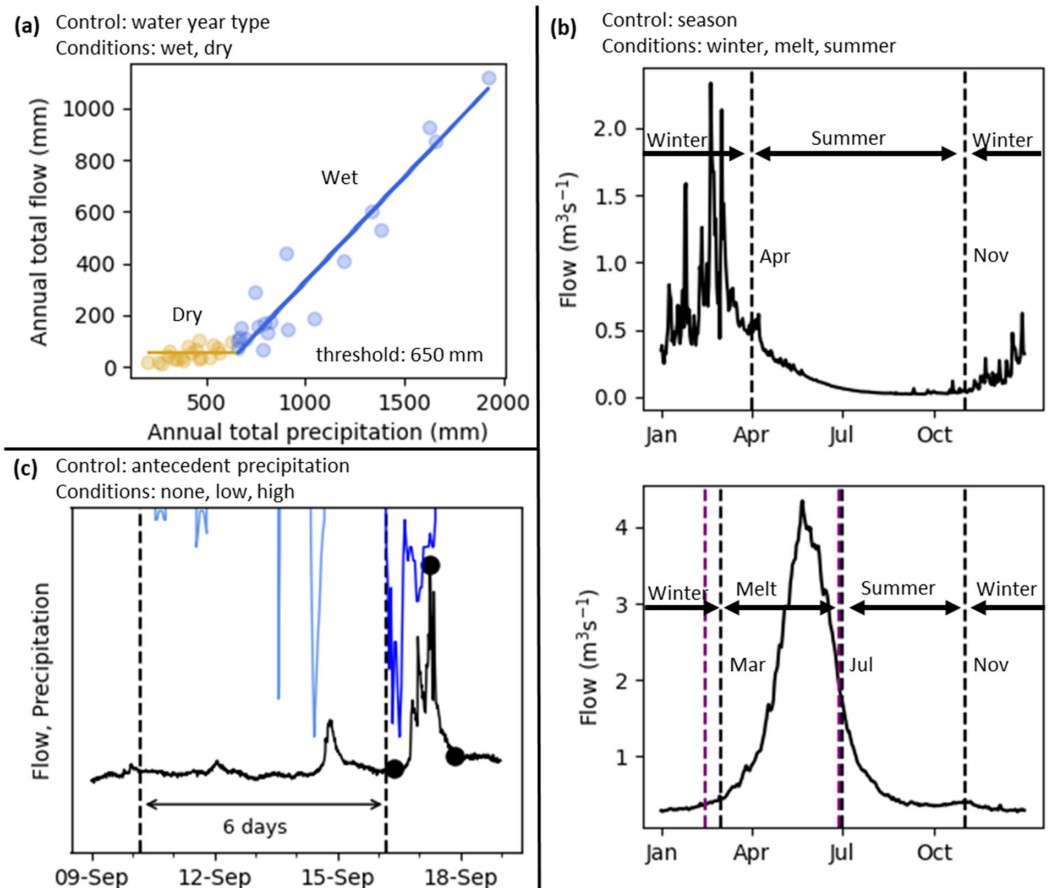

**Figure 4: Example hydrologic condition identification for time-varying hydrologic controls. (a) Water year type wet (blue) and dry (orange) years for Arroyo Seco. The ordinary least squares linear regression lines for above and below the threshold are shown. (b) Seasons (vertical dashed) delineated from the undisturbed average annual hydrograph for a no-snow watershed (top) with winter and summer (Arroyo Seco) and a snow dominated watershed (bottom) with winter, melt, and summer (Clear Creek). The minimum and maximum snow melt dates are shown consecutively (purple dashed). (c) The six-day prior to storm start antecedent precipitation period (between dashed) for an example rainfall-runoff event (rainfall is dark blue, runoff is black). Shown are all storms which are summed within the antecedent precipitation period (light blue).**

### 3.3. Statistical assessment of event-scale hydrologic variability

A number of statistical methods were used to investigate the influence of the time-varying controls and wildfire disturbance on event runoff response. Trends in undisturbed rainfall-runoff patterns were first evaluated using a LOWESS curve. Inferential statistics and the kernel density estimation (KDE) distributions were then used to assess





the effects of time-varying hydrologic conditions on undisturbed rainfall-runoff event metrics. The non-parametric Mann Whitney U Test was used to evaluate the effect of WYT, and the non-parametric Kruskal Wallis Test was used to evaluate the effect of season and antecedent precipitation, all at a 95% confidence level. If significant differences were found based on the Kruskal Wallis Test, the Dunn Test was used to identify specific significant hydrologic conditions. The null hypothesis for all tests was that hydrologic conditions did not impact event metrics.

The statistical test results for all area-normalized metrics were summarized by runoff metric group across and within study watersheds to identify significant hydrologic controls on event runoff response. Summarizing by area-normalized metrics facilitated comparison between watersheds. Summarizing by runoff metric groups facilitated comparison between time-varying hydrologic controls and reduced the emphasis on specific metric calculation methods. The relative importance of each time-varying hydrologic control was assessed for each watershed and runoff

metric group. For each metric group, the significance rate was calculated for the study watersheds together and individually by dividing the number of significant event metrics by the number of metrics in the runoff metric group. When a single hydrologic condition was identified as significant by the Dunn Test, the significant rate for the condition was calculated by dividing the number of significant event metrics for the condition by the number of metrics in the runoff metric group.

Watershed specific significant condition groups were identified for the storm depth and runoff peak relationship in two contrasting watersheds: Arroyo Seco and Clear Creek. The undisturbed rainfall-runoff events in each watershed were sorted into hydrologic condition permutations of the significant hydrologic controls for peak runoff. A power trend was fit to each permutation using ordinary least squares regression. The significant condition groups were identified by combining the permutations with similar power trends. An updated power trend was fit to each significant

condition group.

The influence of the wildfire disturbance on event runoff response was then evaluated within each significant condition group. The percent of post-fire events above the significant group trend and the percent one standard deviation above the trend was calculated for all combined and individual years post-fire. The post-fire percent above the significant group trend and the standard deviation was compared to the expected 50% and 16%, respectively.

**4. Results**

**4.1. RREDI toolkit performance**

The RREDI toolkit performed well across the nine study watersheds and resulted in a rainfall-runoff event dataset of 5042 events (Table S4). 7026 events were initially identified by the RREDI toolkit in step 2. Of these, 774 events (11% of total events, 5 to 34% range across study watersheds) were systematically inspected for event timing and

flagging accuracy (Table 2). Events were identified at a 69% accuracy rate pre-flagging (step 2) and the accuracy rate rose to 90% after flagging (step 4). The identified occurrence rate for each of the four known issues across all watersheds was 2% for 15-minute streamflow data gaps, 13% for diurnal cycling, 4% for duplicate events, and 15% for no identified end time events (Table S5). The total event retention rate after flagging was 72%, with the highest retention rate of 83% in Arroyo Seco and the lowest of 45% in Camp Creek.






**Table 2: RREDI toolkit rainfall-runoff performance results including event accuracy, flagging, and retention rates across the study watersheds.**

| Watershed | Event accuracy pre-flagging (%) | Event accuracy post-flagging (%) | Events retained post-flagging (#) | Events retained post-flagging (%) |
|---|---|---|---|---|
| Arroyo Seco | 88 | 91 | 394 | 83 |
| Ash Canyon Creek | 75 | 78 | 374 | 75 |
| Cache La Poudre | 80 | 93 | 1208 | 72 |
| Camp Creek | 42 | 88 | 162 | 45 |
| Clear Creek | 77 | 89 | 886 | 73 |
| Thompson River | 67 | 91 | 449 | 75 |
| Shitike Creek | 62 | 93 | 663 | 75 |
| Valley Creek | 74 | 91 | 624 | 73 |
| Wet Bottom Creek | 70 | 100 | 282 | 63 |
| Overall | 69 | 90 | 5042 | 72 |

## 4.2. Hydrologic variability

The resulting rainfall-runoff event dataset consisting of 5042 events across the study watersheds allowed for a data-driven analysis of event runoff patterns and controls. The dataset was sufficiently large such that rainfall-runoff events were assigned to all hydrologic conditions in each watershed (Table S6). For undisturbed events across all the study watersheds, there was an increasing trend in runoff peak with increasing storm depth (Figure 5). A slope break was identified at 10 mm storm depth, above which the runoff peak increases more rapidly with increasing storm depth.

Variation between runoff peak and storm depth existed across the watersheds. The identified slope break was most apparent in three watersheds: Arroyo Seco, Shitike Creek, and Wet Bottom Creek. Events above the threshold in the other six watersheds were limited. Four watersheds, Arroyo Seco, Cache La Poudre River, Camp Creek, and Wet Bottom Creek had large spreads in the LOWESS curve residuals compared to the other five watersheds. The remainder of the study results focus on two contrasting watersheds, Clear Creek and Arroyo Seco (Figure 1; Table 1).

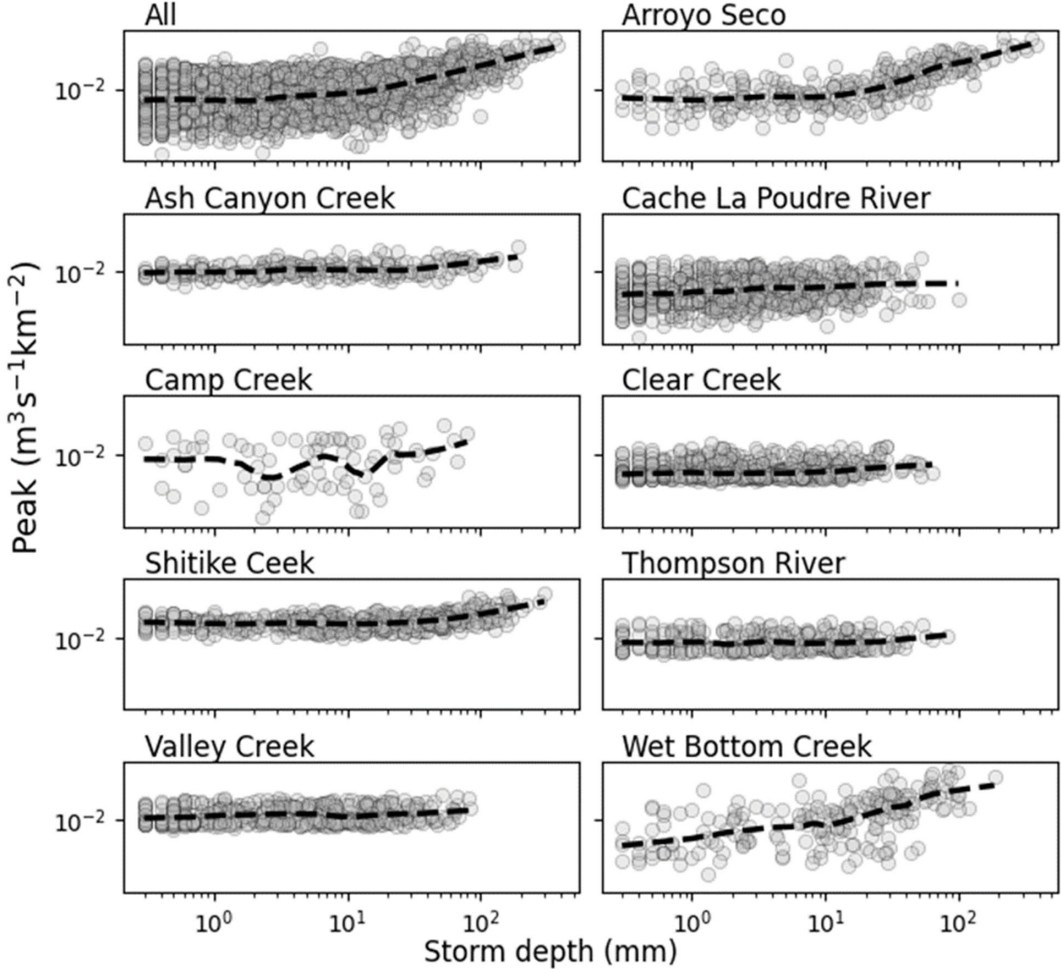


**Figure 5: Undisturbed rainfall-runoff events for storm depth (mm) and runoff peak (m³ s⁻¹ km⁻²). A LOWESS curve (dashed black line) for the undisturbed events for all study watersheds and each individual watershed is shown.**

Clear directional shifts that varied by runoff metric and watershed were apparent in four selected runoff metric undisturbed event distributions for WYT, season, and antecedent precipitation. In both Arroyo Seco and Clear Creek, wet years had a higher median value than dry years for volume, peak, duration, and response time runoff metrics (Figure 6). Winter had higher median values than summer for volume, peak, and response time runoff metrics in Arroyo Seco but shifts were not as consistent in direction in Clear Creek. For antecedent precipitation in both Arroyo

Seco and Clear Creek, the largest median peak runoff and shortest median response time was for high antecedent precipitation.



**Figure 6: Undisturbed rainfall-runoff event KDE distributions for hydrologic conditions for WYT, season, and antecedent precipitation in (a) Arroyo Seco and (b) Clear Creek for four selected runoff metrics: volume, peak, duration, and response time. Distributions are colored by hydrologic condition. The median value of each distribution is shown (dashed line). Significant difference between distributions is indicated (*). Note there is no melt season in Arroyo Seco.**



All time-varying hydrologic controls were found to be significant but significance varied by event runoff metric
and watershed (Figure 6; Table 3). Water year type was the most often significant across the four selected runoff
metrics in Arroyo Seco while season was the most often significant in Clear Creek (Figure 6; Table 3). Antecedent
precipitation was the least significant control in both watersheds and exhibited the most variation in significance
between runoff metrics across the two watersheds. Of the four selected runoff metrics, peak runoff was most
commonly significant across all the study watersheds for all three time-varying hydrologic controls (Tables S7, S8,
S9). Peak runoff was also significant across the three time-varying hydrologic controls for both Arroyo Seco and Clear
Creek except antecedent precipitation in Clear Creek (Figure 6; Table 3). Conversely, the least common significant
metric across all the study watersheds varied by runoff metric, including duration and response time for WYT, duration
for season, and volume for antecedent precipitation (Tables S7, S8, S9). Despite being least commonly significant
overall, response time for WTY in Arroyo Seco and duration for season in Clear Creek were significant (Figure 6;
Table 3).

**Table 3: Undisturbed rainfall-runoff event hydrologic condition statistical test results for the Mann Whitney U Test (WYT) and Kruskal Wallis and Dunn Tests (season, antecedent precipitation) for Arroyo Seco and Clear Creek for four selected runoff event metrics. Shading indicates rejection of the null hypothesis at a**
**significance level of 0.05. In shaded cells, an indicator marks the significantly different condition from the Dunn Test and no indicator means all conditions were significantly different.**

| Watershed | Time-varying hydrologic control | Event metrics | | | |
|---|---|---|---|---|---|
| | | Volume | Peak | Duration | Response time |
| Arroyo Seco | Water year type | <0.001 | <0.001 | 0.05 | 0.005 |
| | Season | 0.48 | 0.013 | 0.15 | 0.47 |
| | Antecedent precipitation | 0.32 | <0.001 + | 0.57 | 0.12 |
| Clear Creek | Water year type | 0.009 | <0.001 | 0.56 | 0.60 |
| | Season | <0.001 * | <0.001 | <0.001 # | <0.001 # |
| | Antecedent precipitation | 0.07 | 0.11 | 0.003 & | 0.008 & |

Seasons: *Winter, ^Melt, #Summer
Antecedent precipitation: &None, ~Low, +High

Water year type and season were generally more important while antecedent precipitation was generally less
important, evaluated by the relative significance rates, across all study watersheds (Figure 7). However, time-varying
hydrologic control importance varied for individual watersheds and area-normalized runoff metric groups. The
watershed-average WYT significance rate exceeded 50% for runoff volume and runoff magnitude metric groups
(Figure 7 a). Water year type was more important than the watershed-average for all metric groups in Arroyo Seco
(Figure 7 b) and for runoff magnitude and runoff volume metrics groups in Clear Creek (Figure 7 c). Water year type
was generally more important than the watershed-average in Arroyo Seco, Ash Canyon Creek, Camp Creek, and
Shitike Creek; less important in Clear Creek, Valley Creek, and Wet Bottom Creek; and similarly important in Cache
La Poudre River and Thompson River (Fig. S8).

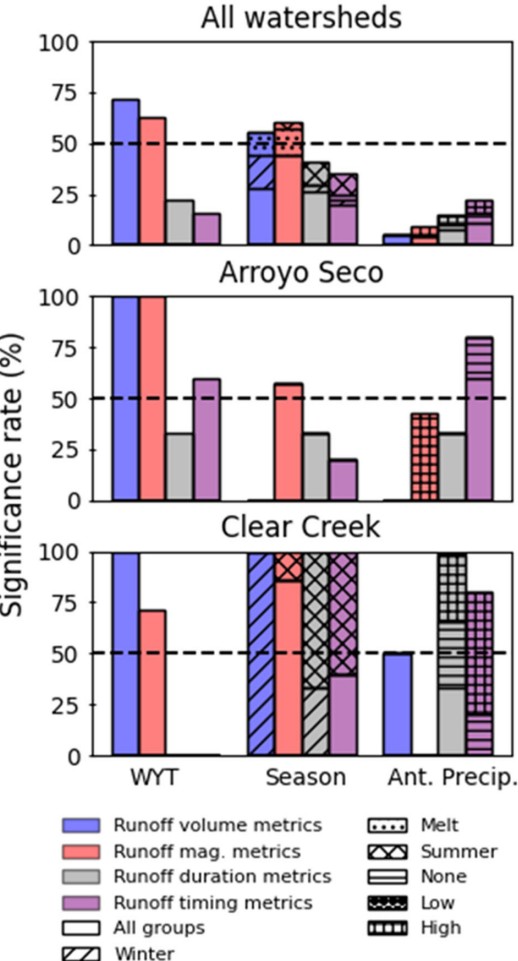

**Figure 7: Area-normalized event runoff metric group significance summary rates for statistical test results. Individual plots for the watershed-average (a), Arroyo Seco (b), and Clear Creek (c). Shown are the average significance rates within the four rainfall-runoff metric groups including runoff volume metrics (blue), runoff magnitude metrics (red), runoff duration metrics (grey), and rainfall-runoff timing metrics (purple). Bars are grouped by time-varying hydrologic control (WYT, season, antecedent precipitation). The WYT group shows results of the Mann Whitney U Test. The season and antecedent precipitation groups show results from the Kruskal Wallis Test. The hatching within the bars represents statistically different individual hydrologic conditions from the Dunn Test where no hatching indicates all hydrologic conditions were statistically different. The 50% rate is highlighted (black dashed).**





The season watershed-average significance rate exceeded 50% for runoff volume and runoff magnitude metric
groups (Figure 7 a). Season was more important than the watershed-average for in no metric groups in Arroyo Seco
(Figure 7 b) and for all metric groups in Clear Creek (Figure 7 c). Season was generally more important than the
watershed-average in Cache La Poudre River, Clear Creek, Thompson River, and Valley Creek; less important in Ash
Canyon Creek and Camp Creek; and similarly important in Arroyo Seco, Shitike Creek, and Wet Bottom Creek (Fig.
S8).

The antecedent precipitation watershed-average significance rate exceeded 50% for no metric groups (Figure 7
a). Antecedent precipitation was more important than the watershed-average for runoff magnitude, runoff duration,
and rainfall-runoff timing metric groups in Arroyo Seco (Figure 7 b) and for runoff volume, runoff duration, and
rainfall-runoff timing metric groups in Clear Creek (Figure 7 c). Antecedent precipitation was generally more
important than the watershed-average in Arroyo Seco and Clear Creek; less important in Ash Canyon Creek, Camp
Creek, Shitike Creek, Thompson River, Valley Creek, and Wet Bottom Creek; and similarly important in Cache La
Poudre River (Fig. S8).

Three and four unique significant condition groups and trends emerged for the storm depth and peak runoff
relationship in Arroyo Seco and Clear Creek, respectively (Figure 8). The watershed specific significant condition
groups were identified from eight and six hydrologic condition permutations of the watershed specific significant
hydrologic controls in Arroyo Seco and Clear Creek, respectively (Fig. S9). The three significant condition groups in
Arroyo Seco were (1) wet none+low, (2) wet high, and (3) dry. The four significant condition groups in Clear Creek
were (1) summer, (2) winter, (3) wet melt, and (4) wet dry. Significant condition group trends were only assessed
above 10 mm storm depth in Arroyo Seco consistent with the storm depth threshold observed in this watershed (Figure
5). Each significant condition group power trend was distinct, falling within a different portion of the all-events
distribution (Figure 8; Table S10).

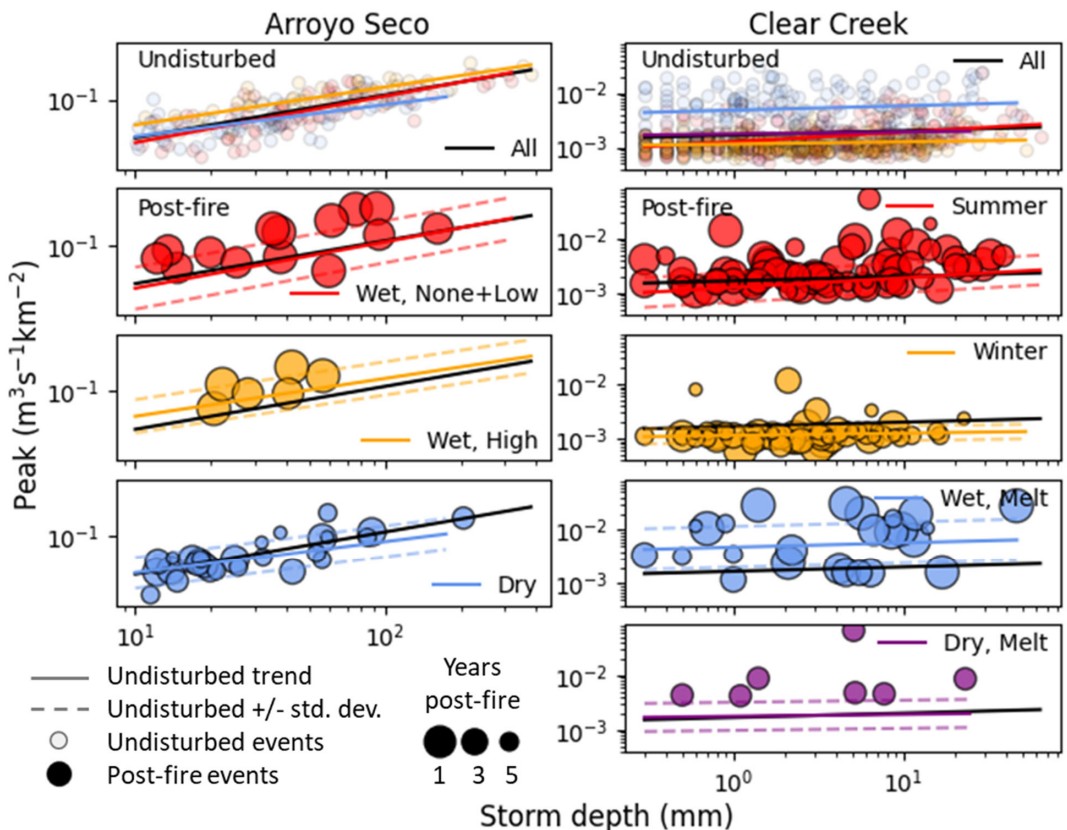

**Figure 8: Significant condition groups for Arroyo Seco and Clear Creek for storm depth (mm) and runoff peak (m³ s⁻¹ km⁻²). Shown are the significant group trends and one standard deviation for each of the significant condition group (colored) and the all-events trend (black). The undisturbed rainfall-runoff events (top) and post-fire events within each significant condition group are shown.**

The portion of post-fire events that fell above the significant condition group trend was generally greater than expected for peak runoff in Arroyo Seco and Clear Creek, however this varied by significant condition group (Figure 8). The percent of post-fire events above the significant condition group trend was at least 50% for all significant condition groups in Arroyo Seco and all groups except winter in Clear Creek (Table S11). The percent of events more than one standard deviation above the significant condition group trend was at least 16% for all significant condition groups except dry in Arroyo Seco and all except winter in Clear Creek. In general, the percent of post-fire events above the significant condition group trend and one standard deviation decreased with increasing time since fire (Figure 8; Table S11).




### 5. Discussion

#### 5.1. RREDI toolkit

The rainfall-runoff event dataset generated by the RREDI toolkit allowed for a large-sample analysis of hydrologic trends and controls across the study watersheds. The toolkit had an overall 90% event accuracy rate, ranging from 78 to 100% across study watersheds. There were no clear physio-climatic patterns to the performance. Lower event accuracy rates in Ash Canyon Creek, Camp Creek, and Clear Creek may be associated with a range of factors including poor quantification of storm timing, water withdrawals, temporally aggregated streamflow, and extended periods of diurnal cycling. The event accuracy increased after removal of flagged events for all study watersheds. Event retention rates were below average in Camp Creek and Wet Bottom Creek but post-flagging event accuracy rates were near average and 100%, respectively. Both watersheds have flashy hydrology and substantial periods of low flow diurnal cycling. This resulted in several identified event pairs where the event runoff response was outside of the allowable response window.

Quantification of storm events influenced the RREDI toolkit performance, where storm timing was a common reason for poor event identification. A gridded precipitation data product was used to overcome sparse rain gage density and limited or sporadic periods of record in the mountainous western US. The rainfall measured in valleys, where long term rain gages are more common (such as the NOAA COOP network), often diverges from mountain rainfall characteristics due to orographic gradients (Roe, 2005). Differences in rain gage distance to the watershed also complicated inter-watershed comparison. Using gridded precipitation allowed for a spatially consistent precipitation time series to be created for all study watersheds. The centroid of the watershed was used here, but future work could incorporate watershed averaged precipitation or other methods to capture precipitation spatial variability (Giani et al., 2022; Kampf et al., 2016; Wang et al., 2023). The high spatial and temporal resolution of the AORC data product performed well compared to rain gage measurements (Hong et al., 2022; Kim & Villarini, 2022). However, the hourly temporal resolution did result in some loss of information related to short duration, high intensity storms as precipitation was linearly interpolated across the timestep.

The RREDI toolkit time-series event separation method was transferable across diverse watersheds using only two watershed specific parameters, and addressed several common issues identified by past studies. The most common rainfall-runoff event separation technique relies on established baseflow methods to isolate event flow (e.g. Chapman & Maxwell, 1996; Duncan, 2019; Eckhardt, 2005; Xie et al., 2020). Runoff events are then identified where baseflow diverges from total flow (Long & Chang, 2022; Mei & Anagnostou, 2015; Merz et al., 2006; Merz & Blöschl, 2009; Tarasova, Basso, Zink, et al., 2018). However, Giana et al., (2022) identified the need for increased method transferability across watersheds. To increase transferability, methods use fewer modifying watershed parameters (Blume et al., 2007; Nagy et al., 2022) or time-series signal processing, as used in the RREDI toolkit, to identify events (Giana et al., 2022; Patterson et al., 2020). A comparison of a baseflow separation method against a time-series signal processing method found good agreement in event identification rates and metrics with the added bonus of transferability in the latter method (Giana et al., 2022). The RREDI toolkit performed best when separating discrete rainfall-runoff events, however with the implementation of the flagging algorithm was able to address issues that have been limiting in other methods. The baseflow separation methods use daily streamflow (Long & Chang, 2022; Mei &



Anagnostou, 2015; Merz et al., 2006; Merz & Blöschl, 2009; Tarasova, Basso, Zink, et al., 2018), however by using 15-minute streamflow the RREDI toolkit could identify and characterize sub-daily events. The use of time-series signal processing also allowed for the identification of events with no runoff response, providing more information about the rainfall thresholds and antecedent conditions required for runoff generation. An algorithm to remove diurnal cycling events was also implemented, something not previously addressed.

The time-series event separation method introduced in this study allowed for large-sample hydrologic analysis to investigate event-scale rainfall-runoff patterns and controls. Future work could expand this analysis to a larger set of watersheds and potential controls (Gupta et al., 2014). The RREDI toolkit could also be applied to address other pressing event-scale hydrologic challenges, including the influence of other watershed disturbances (e.g. urbanization, forest treatments, insect infestation) (Ebel & Mirus, 2014; Goeking & Tarboton, 2020), evaluation of design rainfall events, flood prediction, or event recurrence interval analysis. Beyond rainfall-runoff event analysis, the RREDI toolkit could be used to identify paired events in other rainfall-peaking time-series data relationships such as water quality events (e.g., turbidity) or soil moisture events.

### 5.2. Hydrologic variability

In general, across the study watersheds, WYT and season were significant time-varying hydrologic controls on event runoff response while antecedent precipitation played a lesser role, but significance varied by watershed and runoff metric. Differences in the significance of controls between study watersheds corresponds with the findings of other large-sample rainfall-runoff analysis (Merz et al., 2006; Merz & Blöschl, 2009; Norbiato et al., 2009; Tarasova, Basso, Poncelet, et al., 2018; Tarasova, Basso, Zink, et al., 2018; Wu et al., 2021). Variability in the significance of runoff metrics within a watershed underlined the importance of comparing similar metrics between watersheds and studies to assess event runoff response. Differences between event runoff response in wet and dry years were significant across the runoff metrics in six of the seven watersheds where a WYT precipitation threshold was identified (Figure 7; Fig. S8). This aligns with Biederman et al.'s (2022) finding that the threshold between wet and dry years were important in event runoff response in semi-arid watersheds. Differences in runoff processes in wet and dry years, such as the interaction between soil drainage and vegetation rooting depth as the watershed recovers, may drive these observed differences in runoff response (Bart, 2016; Biederman et al., 2022). High interannual variation in snowpack (Cayan, 1995) may be a driver in WYT significance in six of the seven snow-dominated watersheds. Water year type was significant for one of the two rain dominated watersheds, Arroyo Seco. In Arroyo Seco, extreme variability in the interannual frequency and intensity of atmospheric rivers that bring a majority of the precipitation may explain the WYT significance (Lamjiri et al., 2018). Surprisingly, WYT was not significant in Wet Bottom Creek despite interannual variation in the summer North American Monsoon in this watershed (Arriaga-Ramierez & Cavazos, 2010; Pascolini-Campbell et al., 2015). This may be because despite the monsoon, the majority of precipitation in this watershed instead comes from winter storms (Arriaga-Ramierez & Cavazos, 2010).

Seasonal differences in event runoff response were significant across the runoff metrics in seven watersheds including both snow- and rain-dominated systems (Figure 7; Fig. S8). Similar patterns have been observed across a variety of watersheds with a range of precipitation and streamflow regimes and watershed properties (Merz et al., 2006; Merz & Blöschl, 2009; Norbiato et al., 2009; Tarasova, Basso, Poncelet, et al., 2018). In snow-dominated



watersheds, observed seasonality has been attributed to differences in precipitation type (Merz et al., 2006; Merz &
Blöschl, 2009; Tarasova, Basso, Zink, et al., 2018), seasonal water balance (Berghuijs et al., 2014; Merz et al., 2006;
Tarasova, Basso, Poncelet, et al., 2018), and the influence of snow on antecedent moisture conditions (Hammond &
Kampf, 2020; Merz et al., 2006; Merz & Blöschl, 2009; Norbiato et al., 2009). Seasonality in rain-dominated
watersheds has been attributed to differences in storm properties (intensity, depth) and antecedent moisture driven by
seasonal water balance (Berghuijs et al., 2014; Merz & Blöschl, 2009; Tarasova, Basso, Zink, et al., 2018). In fact,
seasonal water balance has been identified as more important than topography in event runoff response differences
between watersheds (Merz et al., 2006). As storm properties were separately accounted for in this analysis by
evaluating event runoff response with respect to specific storm metrics (e.g. storm depth), the significance of
seasonality is likely associated with seasonal differences in evapotranspiration and soil moisture.

Antecedent precipitation was only significant across the runoff metrics in two very arid watersheds, Arroyo Seco
and Clear Creek (Figure 7; Fig. S8). This finding indicates a complexity in this time varying hydrologic control as
these findings contrast with our expectation that antecedent precipitation, as a proxy for antecedent soil moisture,
would be a control on rainfall-runoff patterns. Antecedent precipitation has been used has a proxy for antecedent soil
moisture in several studies (Long & Chang, 2022; Merz et al., 2006; Tarasova, Basso, Zink, et al., 2018) and in the
SCS curve method for runoff generation (Mishra & Singh, 2003). Past studies have found conflicting results in the
significance of antecedent precipitation. Both 10-day antecedent precipitation in Italy (Merz et al., 2006) and
antecedent soil moisture (Merz & Blöschl, 2009; Tarasova, Basso, Zink, et al., 2018) have been found to influence
event runoff response. However, 10-day antecedent precipitation in Germany (Tarasova, Basso, Zink, et al., 2018)
and 3-day antecedent precipitation in Oregon, US (Long & Chang, 2022) were not significant controls at the event
scale. A possible reason why antecedent precipitation was not identified as significant in seven study watersheds may
be the dominance of the seasonal water balance (Merz et al., 2006) which may not be captured in short window (<10
day) antecedent precipitation (Tarasova, Basso, Zink, et al., 2018). To mitigate this, Tarasova, Basso, Zink, et al.
(2018) suggested applying a longer antecedent precipitation window (30-60 days) to better account for seasonal
changes in the water balance.

In both Arroyo Seco and Clear Creek, significant condition groups revealed distinct trends within the storm depth
and runoff peak relationship (Figure 8). In Arroyo Seco, the runoff peak for a given storm was lower in significant
condition groups with dry condition events than those with wet condition events. Further, with increasing storm depth,
the dry significant condition group trend deviated further below the all-events trend. A possible reason for the
divergence between the wet and dry significant group trends is differences in dominant runoff processes (Bart, 2016;
Biederman et al., 2022) driven by strong interannual variation in wetness conditions (Merz & Blöschl, 2009; Tarasova,
Basso, Zink, et al., 2018). Antecedent precipitation was important during wet years in Arroyo Seco. Interestingly, high
antecedent precipitation mattered more at low storm depths, where the high wet significant condition group trend
returned to the all-events trend with increasing storm depth. This may be due to an increasingly overwhelming
overland runoff response to larger storms that diminishes the influence of antecedent precipitation. Too few events in
the dry significant condition group limited separation of antecedent precipitation so this remains inconclusive.



In Clear Creek, season was the primary driver separating the significant condition groups (Figure 8). This aligns with findings in other snow-dominated watersheds where the seasonal water balance was the primary driver of differences in rainfall-runoff patterns (Merz et al., 2006). This dominance of seasonal water balance over event antecedent precipitation likely explains why antecedent precipitation was not significant in the Clear Creek significant condition groups. Similar to other snow-dominated watersheds, the peak runoff response was highest during the melt and lower in the summer (Merz & Blöschl, 2009). Separation of wet and dry years was only significant during melt, likely due to the dominance of winter precipitation and interannual variance in snowpack in this watershed (Arriaga-Ramierez & Cavazos, 2010; Cayan, 1995). Summer was the most responsive season to increasing storm depth. Without the influence of the snowpack during summer, this responsiveness is consistent with the findings in rain-dominated Arroyo Seco.

Consideration of WYT and seasonality was critical to discerning the influence of wildfire disturbance on event runoff response. The influence of wildfire was most apparent in the winter in Arroyo Seco and summer in Clear Creek (Figure 8). The differences between post-fire response in Arroyo Seco and Clear Creek is consistent with the large range of post-fire responses observed across western US watersheds (Hallema et al., 2017; Saxe et al., 2018). In Arroyo Seco, for each year post-fire the peak runoff events were greater than expected based on the undisturbed event distribution. This post-fire increase in runoff peak is consistent with previously observed increases in total annual flow in the watershed (Bart, 2016; Beyene et al., 2021). In Arroyo Seco, the first two years post-fire were wet years and the subsequent years were dry. Without considering the dry years separately, the influence of the fire would have been obscured within the full undisturbed event distribution. Distilling disturbed event runoff response from natural WYT variability has been identified as a challenge by other studies (Biederman et al., 2022; Hallema et al., 2017; Long & Chang, 2022; Mahat et al., 2016; Owens et al., 2013). Without consideration of WYT, interannual hydrologic variability may obscure changes in post-fire rainfall-runoff patterns (Mahat et al., 2016; Owens et al., 2013) or falsely exaggerate the impact of wildfire if, for example, a fire is followed by very wet years as occurred in Clear Creek.

       Altered post-fire rainfall-runoff patterns also appeared to be seasonal, as observed in Clear Creek (Figure 8). In Clear Creek, post-fire peak runoff was greater than expected every year in summer but the trend was inconsistent in
winter and melt. Biederman et al. (2022) identified a similar trend, greater post-fire change observed in the summer than the winter, in watersheds in the southwest US. Wildfire has also been found to influence snow accumulation and melt timing (Ebel, Hinckley, et al., 2012; Gleason et al., 2019; Kampf et al., 2022; Maina & Siirila-Woodburn, 2020). However, less wildfire influence on event runoff response in the winter and melt in snow-dominated watersheds like Clear Creek makes sense because snow accumulation and melt likely dominate runoff response during these seasons.
The altered post-fire summer events would have been obscured by the larger melt events without considering the seasonality of rainfall-runoff events in Clear Creek. In Oregon, where Long & Chang (2022) found no significant change between pre- and post-fire rainfall-runoff patterns despite comparing two dry years, the seasonality of events may have obscured post-fire impacts as they did in Clear Creek.



## 6. Conclusions

This study presents and utilizes the RREDI toolkit, a novel time-series event separation method, to investigate the influence of time-varying hydrologic controls including WYT, season, and antecedent on wildfire disturbed event runoff response. A rainfall-runoff event dataset, consisting of 5042 events was generated by applying the RREDI toolkit to nine study watersheds in the western US. This dataset was used to investigate rainfall-runoff event patterns, identify significant time-varying hydrologic controls, and evaluate how the identified controls influence event runoff

response in wildfire disturbed watersheds. Results revealed in general, WYT and season were significant time-varying hydrologic controls however significant controls varied across watersheds and runoff metrics. The significance of antecedent precipitation varied across watersheds, indicating a more complex relationship for this control consistent with the literature. Unique trends were identified within significant condition groups in two contrasting watersheds, Arroyo Seco and Clear Creek. Within each of the significant condition groups, the portion of post-fire events that fell

above the significant condition group trend was generally greater than expected for peak runoff. Consideration of these time-varying controls promoted the untangling of wildfire disturbance on event runoff response. This analysis has increased the understanding of controls on rainfall-runoff patterns in undisturbed and disturbed watersheds. This elevates the ability to prepare for watershed management in a future with increasing disturbance regimes.

*Code and Data Availability:* All code for data processing and visualization is available upon request from the author. The RREDI Toolkit python code and documentation for creation of the rainfall-runoff event dataset used in this study can be accessed via hydroshare at https://www.hydroshare.org/resource/797fe26dfefb4d658b8f8bc898b320de/ (Canham & Lane, 2022). Streamflow data from the USGS is publicly available at https://dashboard.waterdata.usgs.gov/ and the AORC precipitation gridded dataset is publicly available at

https://hydrology.nws.noaa.gov/aorc-historic/. Wildfire perimeters are available at https://www.mtbs.gov/ and PRISM gridded precipitation data are available at https://prism.oregonstate.edu/.

*Author Contributions:* HC and BL designed the study. HC performed the analyses with input from BL, CP, and BM. The first draft of the paper was written by HC and reviewed by all co-authors.

*Competing Interests:* The authors declare that they have no conflict of interest.

*Acknowledgements:* The work presented in this manuscript was support by the National Science Foundation (NSF) RAPID grant (award #203212, Lane & Murphy) and the Utah Water Research Laboratory.



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
