# Peer review of "Leveraging a time-series event separation method to untangle time-varying hydrologic controls on streamflow"

_EGUsphere, 2023_

## Author Comment (AC1)

Reviewer Responses to HESS comments re:

**Leveraging a time-series event separation method to untangle time-varying hydrologic controls influence on wildfire disturbance on streamflow**

**The comments from HESS and the reviewers are in black and our responses in blue.

**Contents**

**Reviewer 1**

**General Comments**

The study from Canham et al. explores the time-varying hydrological controls on 5042 rainfall-runoff events from 9 western US watersheds, with aiming to untangle the influence of wildfire on streamflow. The paper is well-crafted, with the supporting data and text well recorded in the supplementary file. However, my main concern is that I think the focus of this paper should be on exploring the influence of wildfire on streamflow. Given that there are already lots of studies focus on rainfall-runoff event separation method or large sample events temporal-spatial controls investigation. Thus, the novelty of this paper should be exploring the wildfire influence on streamflow. Yet, the current paper structure contains large proportion of text describing event separation and also the controls for undisturbed events. So, I think the structure of the paper should be adjusted to highlight your contributions on untangling the wildfire impact on streamflow. My detailed comments can be found below. By the way, I was accidently uploaded my review comments as community comment before. Just ignore the community comment.

    Thank you for taking the time to review and provide comment. We value your suggestions regarding the focus of the paper within the general comment and additionally noted within specific comments 9 and 11.

    Our paper that you reviewed attempted to both perform a large sample hydrologic analysis of time-varying controls with a new event separation method and to assess the wildfire

influence on streamflow. As the reviewer alluded to, this seems to be too much to accomplish in a single paper. However, we respectfully disagree with the reviewer's suggestion that large sample rainfall-runoff events temporal-spatial controls have been well established. A better understanding of hydrologic controls is needed in the context of an increasing disturbance regime (lines 33-40). Furthermore, there remain several identified limitations of existing rainfall-runoff event separation methods highlighted in our paper that the RREDI toolkit seeks to address. To the authors knowledge there has only been a single large sample event analysis evaluating rainfall-runoff controls in the USA as described on lines 61-63, and there is no known western USA specific study. Thus, the authors feel that the RREDI toolkit is a novel event separation method that merits detailed coverage and performance assessment prior to applying it to watershed disturbance analysis. Specifically, the study addresses a need identified by Giani et al. (2022b) to use time-series signal processing to increase transferability across watersheds, and addresses several issues that have been limiting in other studies as described on lines 442-459. Thus, we have re-focused the aims of this paper, as defined on lines 71-74, to: (1) describe and evaluate the performance of a novel time-series event separation method, and (2) apply this method to investigate the influence of time-varying hydrologic controls on event runoff response. Then a second paper will be written to specifically tackle the wildfire influence on streamflow. Therefore, in this revision, we instead *reduced* the focus on wildfire impacts on streamflow in the Introduction, Study Area and Discussion sections and consolidated the application to wildfire disturbances to sections 3.4 *Statistical assessment in wildfire disturbed watersheds*, 4.3 *Hydrologic variability in wildfire disturbed watersheds*, and 5.3 *Hydrologic variability in wildfire disturbed watersheds*. The wildfire disturbed portion of the paper now focuses on two burned case study watersheds leveraging what had been learned in questions 1 and 2 to investigate question 3. There, we briefly demonstrate how accounting for the identified significant time-varying controls could then facilitate an evaluation of the influence of the wildfire disturbance. This sets us up for a second paper focused on wildfire effects on rainfall-runoff patterns, which the reviewer (and the authors) have identified as another novel contribution of this research.

Specifically, in this re-framing of the manuscript, we hypothesize that the observed variability in both rainfall-runoff and post-fire response could be a result of differences in time-varying hydrologic controls including water year type (WYT), season, and antecedent precipitation. To test this, we first evaluated how nine undisturbed study watersheds were influenced by these controls. We found that across the undisturbed watersheds, WYT and season were influential on the event runoff response. We then performed a more in-depth analysis in two burned watersheds, Arroyo Seco and Clear Creek to investigate how these influential controls may have obscured the post-fire rainfall-runoff response influence. We found that antecedent precipitation and seasons, respectively, may have obscured the post-fire streamflow response.

In summary, to address the general comment and specific comments 9 and 11, we have re-worked portions of the paper to separate the post-fire analysis from the large event sample

assessment. We have refocused the analysis of the nine study watersheds on the hydrologic control exploration in undisturbed watersheds (research questions 1 and 2). We have more clearly separated and described the wildfire portions of the analysis in the methods, results, and discussion (created sections 3.4 *Statistical assessment in wildfire disturbed watersheds*, 4.3 *Hydrologic variability in wildfire disturbed watersheds*, and 5.3 *Hydrologic variability in wildfire disturbed watersheds*). We believe that these modifications bring better balance and increase the quality of the work while maintaining each novel portion of the research.

**Specific Comments**

1. Table1: It would be better to add hydrologic characteristics in this table for these catchments, i.e., mean annual precipitation, mean annual potential evapotranspiration, mean annual streamflow and also maybe the streamflow regimes that you mentioned in the line 120-123.

We have included the suggested characteristics including mean annual streamflow, precipitation, and potential evapotranspiration and streamflow regime to Table 1. Additionally, we have removed fire characteristics, see response to general comment.

Table 1: Watershed characteristics for the study watersheds. Where P is precipitation and PET is potential evapotranspiration.

| Watershed | State | USGS Gage ID | Contributing area $(km^2)$ | Streamflow (mean annual) $(m^2\ s^{-1})$ | P (mean annual)* (cm) | PET (mean annual)* (cm) | Streamflow regime |
|---|---|---|---|---|---|---|---|
| Arroyo Seco | CA | 11098000 | 42 | 0.27 | 79 | 777 | Rain |
| Ash Canyon Creek | NV | 10311200 | 14 | 0.10 | 76 | 479 | Snow |
| Cache La Poudre | CO | 06752260 | 2966 | 4.9 | 53 | 449 | Snow |
| Camp Creek | CO | 07103703 | 25 | 0.03 | 56 | 479 | Snow |
| Clear Creek | UT | 10194200 | 426 | 1.0 | 54 | 508 | Snow |
| Shitike Creek | OR | 14092750 | 57 | 2.2 | 157 | 492 | Snow |
| Thompson River | MT | 12389500 | 1652 | 12.1 | 76 | 476 | Snow |
| Valley Creek | ID | 13295000 | 376 | 5.7 | 88 | 401 | Snow |

| Watershed | State | USGS Gage ID | Contributing area (km$^2$) | Streamflow (mean annual) (m$^2$ s$^{-1}$) | P (mean annual)* (cm) | PET (mean annual)* (cm) | Streamflow regime |
|---|---|---|---|---|---|---|---|
| Wet Bottom Creek | AZ | 09508300 | 94 | 0.39 | 62 | 780 | Rain |

*(Falcone, 2011)

2. Line 132: Can you explain what PRISM means?

We have added text to clarify what the PRISM dataset is and updated the citation. PRISM provides many different types of datasets, so we clarified that we used the gridded annual precipitation dataset for the study period in each watershed.

Lines 111-114: "The total annual precipitation at the centroid of each study watershed for each year with available USGS annual streamflow was retrieved from the Parameter-elevation Regressions on Independent Slopes Model (PRISM) gridded annual precipitation dataset (PRISM Climate Group, Oregon State University, 2022)."

PRISM Climate Group, Oregon State University. (2022). *PRISM Climate Data*. https://www.prism.oregonstate.edu/

3. Line 240-242: why it needs to use two different statistical tests to evaluate the effect of WYT and season/antecedent precipitation respectively? In Figure 7, you compared their results in one figure, yet I'm not sure whether the results of these two methods are comparable or not?

We have updated the text to clarify the use of the two tests. The use of each statistical test as used here is appropriate, as the Mann Whitney U test is used to compare two groups while the Kruskal Wallis is used to compare between greater than two groups. The comparison as presented in Fig. 7 is appropriate because there is no direct comparison between results of the two tests as each remains within their respective hydrologic condition. The significance of the two tests were assessed at the same confidence level (line 247).

Lines 245-247: "The non-parametric Mann Whitney U Test was used to evaluate the effect of WYT between the two hydrologic conditions, and the non-parametric Kruskal Wallis Test was used to evaluate the effect of season and antecedent precipitation between three hydrologic conditions, all at a 95% confidence level."

4. Table 2: Does symbol # represent the number of events? If so, please clarify.

We have clarified the text in Table 2 caption as it pertains to this comment.

Lines 296-297: "Table 2: RREDI toolkit performance results including pre- and post-flagging rainfall-runoff event accuracy rates and pre- and post-flagging retention numbers (#) and rates across the study watersheds."

5.  Line 289: How you selected these two contrasting watersheds? The explanation of why you selected these two watersheds as example is needed. Is that possible to compare the results between Arroyo Seco and Valley Creek (this one has similar characteristics with Clear Creek)? Or Maybe Arroyo Seco and Shitike Creek (this one has similar contributing area with Arroyo Seco)? Will the results you observed from Arroyo Seco and Clear Creek also apply to Arroyo Seco and Valley Creek?

To demonstrate the utility of the RREDI toolkit for applications analyzing large hydrologic datasets, we evaluated a suite of undisturbed time-varying hydrologic controls across nine study watersheds, and then performed a more in-depth exploration of watershed disturbance on rainfall-runoff events in two of our study watersheds: Arroyo Seco (CA) and Clear Creek (UT). These watersheds were selected first and foremost because they both experienced wildfires during the period of available streamflow record that burned a significant portion of the watershed (>25%) and with particularly high severity. Additionally, these two case studies provided an interesting comparison with respect to watershed characteristics, as they are an order of magnitude difference in area, are rain vs. snow-melt dominated (respectively), and have a four-fold difference in mean annual streamflow. We utilize the two burned watersheds as a case study for how the investigated hydrologic controls may be obscuring post-fire rainfall-runoff. We expect that the results from this study are transferable to other watersheds, regardless of the burned watersheds selected for the post-fire analysis as we conclude that these controls should be considered when isolating the influence of wildfire on rainfall-runoff patterns (lines 23-25). We have updated the statement within the results noting some results are only presented for the two case study watersheds. Additionally, we have included text within the methods to detail the selection of the two watersheds as case study watersheds for greater in-depth analysis. See response to general comments for more details on this.

Lines 80-100: "Nine study watersheds in the western USA were hand-selected to satisfy a wide range of watershed properties and streamflow regimes from those with streamflow data availability (Fig. 1 a). First, we identified western USA watersheds from the GAGES-II dataset (Falcone, 2011) with at least 20 years of continuous 15-minute streamflow data including at least 10 years of undisturbed streamflow including from wildfire (MTBS, 2023). The selected nine study watersheds spanned a large range of watershed characteristics (Table 1). The contributing areas ranged over three orders of magnitude, from 14 km$^2$ (Ash Canyon Creek) to 2,966 km$^2$ (Cache La Poudre River), with extents defined by the installation locations of the long-term USGS gauges. The mean annual streamflow ranged from 12.1 m$^3$s$^{-1}$ in Thompson River to 0.03

$m^3s^{-1}$ in Camp Creek. The mean annual precipitation ranged from 157 cm in Shitike Creek to 53 cm in Cache La Poudre River (Falcone, 2011) and the mean annual potential evapotranspiration ranged from 780 cm in Wet Bottom Creek and 401 cm in Valley Creek (Falcone, 2011). The watersheds included a range of streamflow regimes including seven snow melt dominated systems with average annual hydrograph peak dates between April and June and two wet season rain dominated systems with average annual hydrograph peak dates between January and February.

Two of the nine study watersheds were selected for a more in-depth exploration of watershed disturbance on rainfall-runoff events: Arroyo Seco and Clear Creek (Fig. 1 b, c). . These watersheds were selected first and foremost because they both experienced wildfires during the period of available streamflow record that burned a significant portion of the watershed (>25%) and with particularly high severity. The Station Fire (2009) burned 100% of Arroyo Seco (78% high and moderate burn severity) and the Twitchell Canyon Fire (2010) burned 25% of Clear Creek (15% high and moderate severity) (MTBS, 2023). Additionally, these two case studies provided an interesting comparison with respect to watershed characteristics, as they are an order of magnitude difference in area, are rain vs. snow-melt dominated respectively, and have a four-fold difference in mean annual streamflow."

Lines 267-270: "Additional statistical methods were performed on two burned study watersheds, Arroyo Seco and Clear Creek, to further explore the influence of wildfire disturbance relative to other time-varying hydrologic controls (Q3; Fig. 2). Arroyo Seco and Clear Creek were contrasting watersheds, with differing watershed characteristics, notably contributing area and streamflow regimes (Table 1) and burn characteristics (Fig. 1 b, c)."

6.  Figure 6: Can you explain what negative values on the x-axis for volume, peak flow and response time mean?

Plotted values are representative of the natural log (ln) of the volume, peak, duration, and response time metrics. As such, values can be negative if less than 1. We have updated the figure so that this is noted appropriately in the x-axis labels. We have clarified this is the natural log transform in the Fig. 6 caption.

[Figure]

Figure 6: Undisturbed rainfall-runoff event KDE distributions for hydrologic conditions for natural log transformed WYT, season, and antecedent precipitation in (a) Arroyo Seco and (b) Clear Creek for four selected runoff metrics: volume, peak, duration, and response time. Distributions are colored by hydrologic condition. The median value of each distribution is shown (dashed line). Significant difference between distributions is indicated (*). Note there is no melt season in Arroyo Seco.

7. Line 329: How do you calculate this relative significance rates?

We have added clarifying text within the methods section to address this confusion.

Lines 253-265: "The statistical test results for all area-normalized metrics were summarized into relative significance rates for each of four runoff metric groups across and within study watersheds to highlight important hydrologic controls on event runoff response. The use of the relative significance rate reduced the issue of multiple comparisons and reduced the emphasis on specific metric calculation methods. Summarizing by area-normalized runoff metrics facilitated comparison between different sized watersheds while summarizing by runoff metric groups facilitated comparison between time-varying hydrologic controls. For each runoff metric group, the significance rate was calculated, either across all study watersheds or for an individual watershed, by dividing the number of significant rainfall-runoff event metrics (based on the Mann Whitney U or Kruskal Wallis test) by the number of metrics in the runoff metric group. When a single hydrologic condition (e.g., melt season) was identified as significant by the Dunn Test, the significance rate for this condition was similarly calculated by dividing the number of significant rainfall-runoff event metrics for the condition by the number of metrics in the runoff metric group. The relative importance of each time-varying hydrologic control was assessed by comparing the significance rates for each watershed and runoff metric group."

Lines 355-259: "For example, in Arroyo Seco, the relative significant rate for the WYT runoff volume metric group was 100%, as two out of the two metrics within this group, runoff volume and runoff ratio (Table S3), were found to be significant by the Mann Whitney U Test (Table S7) while the significance rate for the runoff duration metric group with respect to WYT was 33% because only one out of three metrics was significant. The relative significance rate for the runoff duration with respect to WYT averaged across all nine study watersheds was 72%."

8. Line 350: Can you re-phrase this sentence? It is a bit confused by 'for in no metric groups'.

The paragraphs and figure 7 caption discussing the significance rates have been clarified to refer to differentiating of runoff event metric values across study watersheds with respect to hydrologic controls and the identified sentence grammar has been corrected.

For example, on lines 379-382: "In Arroyo Seco, no runoff metric groups were better differentiated with respect to season than the average significance across all watersheds (Fig. 7 b). Conversely, all runoff metric groups in Clear Creek were better differentiated with respect to season than across all watersheds (Fig. 7 c)."

9. The results section contains a large proportion of analysis on undisturbed rainfall-runoff events, while the analysis of wildfire impacts on streamflow is not sufficiently thorough. Only examples from two watersheds were presented. The focus of the paper should be on

wildfire disturbed streamflow. Adjustment of results proportions and focus of analysis is needed.

We have separated the wildfire portion of the results into a single, smaller section, "4.3 *Hydrologic variability in wildfire disturbed watersheds*". We think that this clarifies how RREDI and time-varying hydrologic controls can be used to assess wildfire disturbed streamflow patterns and brings balance to the results. Additionally, please see response to the general comment.

10. Discussions with more recent large sample rainfall-runoff events controls analysis should be added, i.e. Jahanshahi and Booij (2024) https://doi.org/10.1080/02626667.2024.2302420, Zheng et al. (2023) https://doi.org/10.1029/2022WR033226.

We highlight that there are a number of studies with large sample rainfall-runoff events that we have mentioned and cited in the paper. We appreciate the reviewer bringing our attention to these more recent additions to this body of literature, and we have added additional discussion regarding Jahanshahi and Booij (2024) and have added both to our general comments about these types of studies throughout the paper. We do not feel that any additional commentary above what has already been stated about these types of studies is necessary in the context of the present study.

Lines 507-510: "Past studies have found conflicting results in the significance of antecedent precipitation. Both 10-day antecedent precipitation (Merz et al., 2006) and antecedent soil moisture in Italy (Merz & Blöschl, 2009; Tarasova et al., 2018b) and 5-day antecedent precipitation in Iran (Jahanshahi and Booij, 2024) have been found to influence event runoff response.

Jahanshahi and Booij (2024) included on lines 48, 52, 54, 472, 490, 495, 497, 506, 513.

Zheng et al. (2023) included on lines 49, 51, 53, 61, 473, 491.

Jahanshahi A., Booij M. J. (2024). Flood process types and runoff coefficient variability in climatic regions of Iran. *Hydrological Sciences Journal*, 69:2, 241-258. https://doi.org/10.1080/02626667.2024.2302420

Zheng, Y., Coxon, G., Woods, R., Li, J., Feng, P. (2023). Controls on the Spatial and Temporal Patterns of Rainfall-Runoff Event Characteristics - A Large Sample of Catchments Across Great Britian. *Water Resources Research*, 59. https://doi.org/10.1029/2022WR033226.

11. In the discussion section, it should also have a separate subtitle and section focus more on the impact of wildfire to streamflow.

This comment appears to reflect the reviewer's general comment above. We have separated the wildfire portion of the discussion out to a separate section, "5.3 *Hydrologic variability in wildfire disturbed watersheds*". Additionally, we streamlined the "5.2 *Hydrologic variability*" section of the discussion to focus on the nine watersheds. For more detail, please reference our response to the general comment above.

---

## Author Comment (AC2)

Reviewer Responses to HESS comments re:

**Leveraging a time-series event separation method to untangle time-varying hydrologic controls influence on wildfire disturbance on streamflow**

**\*\*The comments from HESS and the reviewers are in black and our responses in blue.**

**Contents**

**Reviewer 2**

**General Comments**

The writing and methods are generally ok, and this manuscript seems prima facie like it might be appropriate for publications assuming the below comments are sufficiently addressed. With that said, there are a number of issues that are making the methods used difficult to understand and by extension, difficult for readers/reviewers to gauge the appropriateness of the methods used. Most notably, there is a significant amount of subjective interpretation that could easily be replaced with statistical methods.

Thank you for taking the time to review and provide comment. We value your suggestions regarding the descriptions, appropriateness, and repeatability of the methods within this paper. We have worked to address the general and specific comments and have provided a response for each. We believe these comments improve the quality of the work.

1. Isn't a watershed an imprecise concept? You can look at an individual subdrainage which is part of a drainage which is part of a watershed which might be part of a larger watershed complex. However, the "Study Area" section does not acknowledge this when describing how the watersheds were selected. It would be helpful to address this uncertainty and *briefly* justify why a certain selection approach was used. was used.

We have updated the text in Section 2: *Study Watersheds* to provide more detail about the study watersheds based on feedback from both reviewers. We identified nine study watersheds

that a) spanned a broad range of watershed characteristics (Table 1) and b) had a minimum of 20 years of continuous 15-minute streamflow data, including at least 10 years of undisturbed streamflow. The watershed extents were precisely defined by the upstream area contributing flow to the historic streamflow gages. Finally, c) watersheds that experienced a wildfire disturbance during the period of record. Selection criteria c, the burned watersheds, was added to aid in future work. However, in this study, we chose to focus on the undisturbed signals from the nine study watersheds and post-fire signals from two case study burned watersheds.

Lines 80-100: "Nine study watersheds in the western USA were hand-selected to satisfy a wide range of watershed properties and streamflow regimes from those with streamflow data availability (Fig. 1 a). First, we identified western USA watersheds from the GAGES-II dataset (Falcone, 2011) with at least 20 years of continuous 15-minute streamflow data including at least 10 years of undisturbed streamflow including from wildfire (MTBS, 2023). The selected nine study watersheds spanned a large range of watershed characteristics (Table 1). The contributing areas ranged over three orders of magnitude, from 14 km$^2$ (Ash Canyon Creek) to 2,966 km$^2$ (Cache La Poudre River), with extents defined by the installation locations of the long-term USGS gauges. The mean annual streamflow ranged from 12.1 m$^3$s$^{-1}$ in Thompson River to 0.03 m$^3$s$^{-1}$ in Camp Creek. The mean annual precipitation ranged from 157 cm in Shitike Creek to 53 cm in Cache La Poudre River (Falcone, 2011) and the mean annual potential evapotranspiration ranged from 780 cm in Wet Bottom Creek and 401 cm in Valley Creek (Falcone, 2011). The watersheds included a range of streamflow regimes including seven snow melt dominated systems with average annual hydrograph peak dates between April and June and two wet season rain dominated systems with average annual hydrograph peak dates between January and February.

Two of the nine study watersheds were selected for a more in-depth exploration of watershed disturbance on rainfall-runoff events: Arroyo Seco and Clear Creek (Fig. 1 b, c). . These watersheds were selected first and foremost because they both experienced wildfires during the period of available streamflow record that burned a significant portion of the watershed (>25%) and with particularly high severity. The Station Fire (2009) burned 100% of Arroyo Seco (78% high and moderate burn severity) and the Twitchell Canyon Fire (2010) burned 25% of Clear Creek (15% high and moderate severity) (MTBS, 2023). Additionally, these two case studies provided an interesting comparison with respect to watershed characteristics, as they are an order of magnitude difference in area, are rain vs. snow-melt dominated respectively, and have a four-fold difference in mean annual streamflow."

2. Some of the data processing steps are unclear. Particularly regarding what conversions/algorithms were applied to the precipitation data and why they were used.

We note that the reviewer had numerous comments about unclear methodological steps, specifically comments 4, 5, 7, 9, 12, 13, 14, 15, 16, 17, 19, and 20. We have now addressed each and we feel this has greatly improved the clarity of the data processing methods. We have additionally restructured the methods section to better break apart the statistical assessment of event-scale hydrologic variability (section 3.3 *Statistical assessment of event scale hydrologic variability*) and statistical assessment in wildfire disturbed watersheds (section 3.4 *Statistical assessment in wildfire disturbed watersheds*). Additionally, the Supplemental Information (SI) includes a documented description of the RREDI toolkit methods and parameters used for this study and for each watershed. We have included text to clarify this information is available within the SI.

Lines 126-128: "We describe the four key steps of the RREDI toolkit in section .3.1 (Fig. 2) with additional in-depth details in Supplemental Information (SI) section 1. *RREDI toolkit*. A rainfall-runoff event dataset, available in SI (Table S4), was created by applying the RREDI toolkit to nine western USA watersheds (Fig. 2)."

Particularly regarding the processing steps applied to the precipitation data, we clarified the text within section 2.1 *Hydrologic Inputs* detailing the pre-processing of the precipitation data and within section 3.1 *RREDI toolkit* to clarify the rainfall metrics used within the RREDI toolkit,

Lines 114-122: "Hourly precipitation time series were obtained for the watershed centroid from the Analysis of Record Calibration (AORC) 4km$^2$ resolution data product for water years 1980 to 2022 (Fall et al., 2023; National Weather Service Office of Water Prediction, 2021). Linear interpolation was used to develop an instantaneous precipitation record at the AORC resolution of 1 mm by identifying uniform sub-timesteps within the hour timestep resolution. For example, hourly precipitation of 2 mm depth was uniformly spread over the hour with two timestamps of 1mm each. The AORC data product was selected because of comparable or higher correlation between the AORC data product and rain gage measurements compared to other gridded precipitation data products in studies in a mountainous area in Colorado, USA, Louisiana, USA, and the Great Lakes basins (Hong et al., 2022; Kim & Villarini, 2022, Partridge et al., 2024)."

Line 155-159: "In step 1 of the RREDI toolkit, rainfall-runoff event pairs and the associated event window were identified using daily streamflow and precipitation data based on the co-occurrence of separately identified rainfall events by separating precipitation time-series into rainfall and runoff events using signal processing theory from the overlapping period of record (Fig. 2). Rainfall events were characterized by the duration, depth, and 60-minute intensity."

3. The fourth step of the RREDI algorithm is concerning. Why couldn't these events be removed in an earlier step?

Steps 1-3 of RREDI aimed to identify rainfall-runoff event pairs and calculate associated rainfall-runoff event metrics. We then expected and identified several instances of commonly incorrectly identified rainfall-runoff events at the end of step 3, including rainfall-runoff events with: gaps in 15-minute streamflow data, diurnal cycling identified by regular daily rises and falls of flow commonly due to irrigation or snow melt cycles, duplicate rainfall-runoff events, and no identified runoff event end time. These misidentified rainfall-runoff events from a time-series analysis perspective were very similar in appearance to rainfall-runoff events, however were functionally driven by different or uncertain processes that were not applicable to the application of the RREDI toolkit. A good example of this was diurnal cycling, where daily rises in flow driven by irrigation or diurnal snow melt cycles when co-occurring with a rainfall event were often similar in appearance to a rainfall-runoff event. However, these events were driven by a functionally different process. These misidentified events were not representative of rainfall-runoff events and therefore should be removed from a rainfall-runoff event dataset. As a result, step 4 of RREDI was specifically developed to flag and remove these misidentified events from the final dataset of rainfall-runoff events. We have identified and described in detail each of the four common instances of incorrect rainfall-runoff event identification (lines 170-175, Fig. S3) and provided a discussion detailing the implementation of the RREDI toolkit step 4 that effectively "was able to address issues that have been limiting in other methods" (lines 452-454).

Lines 170-175: "Finally, in step 4, event flagging was performed to remove incorrectly identified rainfall-runoff events falling within four event identification issues: gaps in 15-minute streamflow data, diurnal cycling identified by regular daily rises and falls of flow commonly due to irrigation or snow melt cycles (Fig. S5), duplicate rainfall-runoff events, and no identified runoff event end time (Fig. 2; Fig. S3). From a time-series analysis perspective, these misidentified rainfall-runoff events were very similar in appearance to true rainfall-runoff events but were functionally driven by different or uncertain processes that were not applicable to the application of the RREDI toolkit and thus removed."

Lines 442-459: "The RREDI toolkit time-series event separation method is transferable across diverse watersheds using only two watershed specific parameters, and addresses several common issues identified by past studies. The most common rainfall-runoff event separation technique relies on established baseflow methods to isolate event flow (e.g. Chapman & Maxwell, 1996; Duncan, 2019; Eckhardt, 2005; Xie et al., 2020). Runoff events are then identified where baseflow diverges from total flow (Long & Chang, 2022; Mei & Anagnostou, 2015; Merz et al., 2006; Merz & Blöschl, 2009; Tarasova et al., 2018b). However, Giani et al., (2022b) identified the need for increased method transferability across watersheds. To increase transferability, methods use fewer modifying watershed parameters (Blume et al., 2007; Nagy et al., 2022) or time-series signal processing, as used in the RREDI toolkit, to identify rainfall-runoff events (Giania et al., 2022b; Patterson et al., 2020). A comparison of a baseflow separation method against a time-series signal processing method found good agreement in rainfall-runoff event identification rates and metrics with the bonus of transferability in the latter method (Giani et al.,

2022b). The RREDI toolkit performed best when separating discrete rainfall-runoff events, however with the implementation of the flagging algorithm was able to address issues that have been limiting in other methods. The baseflow separation methods use daily streamflow (Long & Chang, 2022; Mei & Anagnostou, 2015; Merz et al., 2006; Merz & Blöschl, 2009; Tarasova et al., 2018b), however by using 15-minute streamflow the RREDI toolkit could identify and characterize sub-daily rainfall-runoff events. The use of time-series signal processing also allowed for the identification of rainfall events with no runoff response, providing more information about the rainfall thresholds and antecedent conditions required for runoff generation. An algorithm to remove diurnal cycling events was also implemented, something not previously addressed."

4. The methods make heavy use of visual interpretation to validate and set thresholds. A more quantitative/deterministic approach is highly recommended. Example: "Total annual streamflow and precipitation were plotted for the undisturbed period of record to visually identify the annual precipitation threshold above which streamflow increased linearly with precipitation." Additionally, "Winter, melt, and summer hydrologic seasons were identified for each watershed based on inspection of the average annual hydrograph". There are methods available for specifically handling these kind of problems. See change point analysis.

Thank you for raising this issue of study repeatability, which is something we thought a lot about in the development of our study methods. Our study objectives were to first describe and evaluate the RREDI toolkit and second, apply RREDI to investigate the influence of time-varying hydrologic controls (WYT, season, and antecedent precipitation) on event runoff response (lines 71-74). It is important to note that we do not expect that using other methods (e.g. change point detection) to adjust the used thresholds in this study to re-assign rainfall-runoff events would substantially alter our proposed approach or findings (added in lines 226-228). Our primary conclusion for objective 2 was that "consideration of the time-varying hydrologic controls, particularly water year type and season, were identified as important when isolating the influence of wildfire on the rainfall-runoff patterns" (lines 23-25). In this manner, we suggest that the evaluated time-varying controls should be considered in future research, while acknowledging that the influence of these controls may be varied based on the method and thresholds used to assign rainfall-runoff events to conditions and the watershed of interest.

Regarding our first objective, "given the inherent challenges of deterministically identifying rainfall-runoff events from only streamflow and precipitation data, we took a time-series signal processing approach that relies in part on expert understanding to define "accurate" rainfall-runoff events like numerous other large-sample hydrology studies including Patterson et al. (2020), Tarasova et al (2018b), and Giani et al. (2022b). Additional in-depth descriptions of each step are included in SI section 1. RREDI toolkit (Fig. S1-S5). All watershed specific and calibrated parameters used are also documented (Table S1, S2)." (added in lines 146-151). Along

with the RREDI toolkit algorithm being available on Hydroshare, Canham & Lane (2022), and analysis of publicly available data, all results are repeatable. We have clarified the text within section 3. *Methods* to note that specific analysis parameters are documented in the SI (see response to general comment 2).

With respect to our second objective, to the authors' knowledge there is no one correct or truly objective way to distinguish water year types and seasons. "Instead of developing entirely new methods to define water year type or season across many watersheds with different hydrologic settings, [which is an interesting but different research aim], we chose to provide sufficient details on methods and results of our expert-informed selections to support a robust, transparent assessment of these time-varying hydrologic variables on event runoff response" (added in lines 202-205). For instance, we could have used percentiles of annual precipitation to more objectively and repeatably assign water year types, but this approach does not account for underlying hydrologic process differences. Instead, water year type was assigned following methods established in Biederman et al. (2022) by identifying the pronounced breakpoint in annual streamflow-precipitation plots indicative of shifts from baseflow to precipitation driven discharge. "Alternative methods such as change point detection may be able to more objectively identify that breakpoint, but automating water year or season identification was beyond the scope of our study" (added in lines 210-213). Similarly, there is not a single objective approach for distinguishing hydrologic seasons across watersheds with the available data (streamflow and precipitation time-series) given the complex physical processes underlying these seasonal changes. We opted to use a combination of visual interpretation of the flow regime and high-resolution spatial snowmelt data and clearly describe our decision-making process and resulting thresholds in section 3.2 *Hydrologic condition identification and assignment* to prioritize isolating the influence of rainfall and soil moisture on runoff rather than snowmelt and rain-on-snow influences. Most importantly, we reiterate and added into the main document (lines 2226-228) that we do not expect that using other methods (e.g. change point detection) to adjust water year type or season thresholds would substantially alter our proposed approach or findings.

5. The problem of multiple comparisons should be addressed here. There are a large number of statistical tests being performed and there is a risk that some of the detected relationships are spurious.

Thank you for raising the issue of multiple comparisons, this is something we have thought a lot about in the development of our study statistical analysis methods. You are correct in that we performed many statistical tests between hydrologic control condition groups for many metrics using a large dataset of rainfall-runoff events. The use of statistical tests here does not imply causation, they simply say that the tested groups are or are not likely to be the same when separated by this hydrologic control group. To get away from possible individual spurious results, including results from the specific calculation of a particular rainfall-runoff metric, we developed the significance rate as detailed in section 3.3 *Statistical assessment of event-scale hydrologic variability* (lines 253-265). The significance rate used to summarize the test results

facilitated the comparison between hydrologic controls and watersheds. We highlight key statistical test results within the main document Table 3 and all statistical test results are available in the Supplemental Information Tables S7-S9. Additionally, Fig. 7 serves to summarize the statistical analyses and the specific results for each watershed are available in SI Fig. S8. We have included a statement within the methods clarifying the development and use of the significance rates as a method to address the issue of multiple comparisons. Additionally, see response to general comment 6.

Lines 253-265: "The statistical test results for all area-normalized metrics were summarized into relative significance rates for each of four runoff metric groups across and within study watersheds to highlight important hydrologic controls on event runoff response. The use of the relative significance rate reduced the issue of multiple comparisons and reduced the emphasis on specific metric calculation methods. Summarizing by area-normalized runoff metrics facilitated comparison between different sized watersheds while summarizing by runoff metric groups facilitated comparison between time-varying hydrologic controls. For each runoff metric group, the significance rate was calculated, either across all study watersheds or for an individual watershed, by dividing the number of significant rainfall-runoff event metrics (based on the Mann Whitney U or Kruskal Wallis test) by the number of metrics in the runoff metric group. When a single hydrologic condition (e.g., melt season) was identified as significant by the Dunn Test, the significance rate for this condition was similarly calculated by dividing the number of significant rainfall-runoff event metrics for the condition by the number of metrics in the runoff metric group. The relative importance of each time-varying hydrologic control was assessed by comparing the significance rates for each watershed and runoff metric group."

Lines 355-359: "For example, in Arroyo Seco, the relative significant rate for the WYT runoff volume metric group was 100%, as two out of the two metrics within this group, runoff volume and runoff ratio (Table S3), were found to be significant by the Mann Whitney U Test (Table S7) while the significance rate for the runoff duration metric group with respect to WYT was 33% because only one out of three metrics was significant. The relative significance rate for the runoff duration with respect to WYT averaged across all nine study watersheds was 72%."

We additionally thought a lot about ensuring that comparable group sizes were evaluated in the statistical tests. The proportion of rainfall-runoff events within each of the evaluation conditions is identified in Supplemental Information Table S6. The group sizes for all groups were determined to be acceptable within the use of the statistical methods applied within this study. Some of the more imbalanced group sizes for the antecedent precipitation conditions may have contributed to the inconsistent results for this control. We have included text within the discussion to acknowledge this.

Lines 301-303: "The rainfall-runoff event dataset was sufficiently large that the proportion of rainfall-runoff events in the hydrologic conditions for each watershed should allow for the use of the described inferential statistical methods (Table S6)."

We also explored the ability to perform similar statistical tests between the post-fire rainfall-runoff events and the undisturbed significant condition groups, however we determined that the group sizes were not comparable due to insufficient number of post-fire rainfall-runoff events. As a result, for the wildfire portion of the study, we developed the statistical methods as detailed in section 3.4. *Statistical assessment in wildfire disturbed watersheds*.

6. There is also a need to make sure that "significant" relationships are of "practical significance". You seem to be working with large datasets that may easily yield detectable differences/relationships, but may only have negligible effect sizes.

We have calculated the effect sizes for each statistically significant result, which allows the reader to further assess the results. Additionally, see response to general comment 5.

Lines 250-252: "The effect size for each significant test result was calculated using the Glass biserial rank correlation coefficient for the Mann Whitney U Test results and the Eta squared test for the Kruskal Wallis Test results (Tables S7, S8, S9)."

Effect size values are included within tables S7, S9 and S10.

**Table S7: Undisturbed rainfall-runoff events Mann Whitney U Test p-value results for wet and dry conditions for the water year type time-varying hydrologic control. Shading indicates rejection of the null hypothesis at a significance level of 0.05. Glass biserial rank correlation values as an effect size indicator are in parenthesis for significant results.**

| Rainfall-runoff event metric | Arroyo Seco | Ash Canyon Creek | Cache La Poudre River | Camp Creek | Clear Creek | Shitike Creek | Thompson River | Valley Creek | Wet Bottom Creek |
|---|---|---|---|---|---|---|---|---|---|
| volume | <0.001 (0.33) | 0.003 (0.29) | 0.02 (0.10) | 0.005 (0.42) | 0.009 (0.14) | <0.001 (0.26) | 0.02 (0.17) | 0.81 | 0.10 |
| RR | <0.001 (0.34) | <0.001 (0.37) | 0.27 | <0.001 (0.55) | 0.003 (0.16) | 0.02 (0.17) | 0.12 | 0.19 | 0.05 |
| peak | <0.001 (0.44) | 0.006 (0.27) | <0.001 (0.26) | <0.001 (0.59) | <0.001 (0.30) | <0.001 (0.59) | <0.001 (0.34) | 0.002 (0.19) | 0.16 |
| Rise | <0.001 (0.34) | 0.02 (0.23) | 0.004 (0.12) | 0.14 | 0.002 (0.17) | <0.001 (0.32) | 0.02 (0.17) | 0.43 | 0.28 |
| Fall | <0.001 (0.40) | 0.03 (0.21) | <0.001 (0.18) | 0.02 (0.34) | <0.001 (0.19) | <0.001 (0.43) | <0.001 (0.25) | 0.59 | 0.27 |
| Rise_percent | <0.001 (0.24) | 0.07 | 0.14 | 0.37 | 0.64 | 0.007 (0.20) | 0.23 | 0.19 | 0.71 |
| Fall_percent | <0.001 (0.24) | 0.22 | 0.02 (-0.09) | 0.15 | 0.64 | <0.001 (0.26) | 0.10 | 0.04 (-0.13) | 0.57 |
| RiseRate | <0.001 (0.34) | 0.71 | 0.01 (0.11) | 0.35 | 0.02 (0.12) | <0.001 (0.30) | 0.04 (0.15) | 0.70 | 0.65 |
| FallRate | <0.001 (0.45) | 0.10 | <0.001 (0.13) | 0.19 | <0.001 (0.19) | <0.001 (0.45) | 0.011 (0.19) | 0.37 | 0.36 |
| duration | 0.05 | 0.10 | 0.09 | 0.04 (0.30) | 0.56 | 0.24 | 0.08 | 0.50 | 0.07 |
| rising_dur | 0.008 | 0.05 | 0.44 | 0.82 | 0.19 | 0.03 | 0.10 | 0.08 | 0.04 |

**RC2 HESS Review Responses**

| | | | | | | | | | |
|---|---|---|---|---|---|---|---|---|---|
| falling_dur | 0.48 (0.17) | 0.99 | 0.20 | 0.012 (0.37) | 0.48 | 0.50 (0.16) | 0.12 | 0.58 | 0.36 (0.29) |
| RT1 | 0.005 (0.19) | 0.13 | 0.13 | 0.85 | 0.60 | 0.14 | 0.06 | 0.62 | 0.56 |
| RT2 | 0.50 | 0.41 | 0.02 (-0.10) | 0.53 | 0.34 | 0.006 (-0.21) | 0.2 | 0.013 (-0.15) | 0.23 |
| T2P1 | <0.001 (0.26) | 0.40 | 0.90 | 0.78 | 0.14 | 0.13 | 0.05 | 0.04 (0.13) | 0.09 |
| T2P2 | 0.002 (0.20) | 0.21 | 0.61 | 0.58 | 0.07 | 0.91 | 0.55 | 0.47 | 0.11 |
| T2P3 | 0.93 | 0.20 | 0.36 | 0.48 | 0.06 | 0.30 | 0.62 | 0.39 | 0.63 |
| volume_area | <0.001 (0.33) | 0.003 (0.29) | 0.02 (0.10) | 0.005 (0.42) | 0.009 (0.14) | <0.001 (0.26) | 0.02 (0.17) | 0.81 | 0.10 |
| RR_area | <0.001 (0.34) | <0.001 (0.37) | 0.27 | <0.001 (0.55) | 0.003 (0.16) | 0.02 (0.17) | 0.12 | 0.19 | 0.05 |
| Peak_area | <0.001 (0.44) | 0.006 (0.27) | <0.001 (0.26) | <0.001 (0.59) | <0.001 (0.30) | <0.001 (0.59) | <0.001 (0.34) | 0.00 (0.19) | 0.16 |
| Rise_area | <0.001 (0.34) | 0.02 (0.23) | 0.004 (0.12) | 0.14 | 0.002 (0.17) | <0.001 (0.32) | 0.02 (0.17) | 0.43 | 0.28 |
| Fall_area | <0.001 (0.40) | 0.03 (0.21) | <0.001 (0.18) | 0.02 (0.34) | <0.001 (0.19) | <0.001 (0.43) | <0.001 (0.25) | 0.59 | 0.27 |
| Rise_percent_area | <0.001 (0.24) | 0.07 | 0.14 | 0.37 | 0.64 | 0.007 (0.20) | 0.23 | 0.19 | 0.71 |
| Fall_percent_area | <0.001 (0.24) | 0.22 | 0.02 (-0.09) | 0.15 | 0.64 | <0.001 (0.26) | 0.1 | 0.04 (-0.13) | 0.57 |
| RiseRate_area | <0.001 | 0.02 | 0.00 | 0.14 | 0.002 | <0.001 | 0.02 | 0.43 | 0.28 |

**RC2 HESS Review Responses**

|  |  |  |  |  |  |  |  |  |  |
|---|---|---|---|---|---|---|---|---|---|
|  | (0.34) | (0.23) | (-0.12) |  | (0.17) | (0.32) | (0.17) |  |  |
| FallRate_area | <0.001 (0.40) | 0.03 (0.21) | <0.001 (0.18) | 0.02 (0.34) | <0.001 (0.19) | <0.001 (0.43) | <0.001 (0.25) | 0.59 | 0.27 |
| duration_area | 0.05 | 0.10 | 0.09 | 0.04 (0.30) | 0.56 | 0.24 | 0.08 | 0.50 | 0.07 |
| rising_dur_area | 0.008 (0.17) | 0.05 | 0.44 | 0.82 | 0.19 | 0.03 (0.16) | 0.10 | 0.08 | 0.04 (0.29) |
| falling_dur_area | 0.48 | 0.99 | 0.20 | 0.012 (0.37) | 0.48 | 0.50 | 0.12 | 0.58 | 0.36 |
| RT1_area | 0.005 (0.19) | 0.13 | 0.13 | 0.85 | 0.6 | 0.14 | 0.06 | 0.62 | 0.56 |
| RT2_area | 0.50 | 0.41 | 0.02 (-0.10) | 0.53 | 0.34 | 0.006 (-0.21) | 0.20 | 0.013 (-0.15) | 0.23 |
| T2P1_area | <0.001 (0.26) | 0.40 | 0.90 | 0.78 | 0.14 | 0.13 | 0.05 | 0.04 (0.13) | 0.09 |
| T2P2_area | 0.002 (0.20) | 0.21 | 0.61 | 0.58 | 0.07 | 0.91 | 0.55 | 0.47 | 0.11 |
| T2P3_area | 0.93 | 0.20 | 0.36 | 0.48 | 0.06 | 0.30 | 0.62 | 0.39 | 0.63 |

**Table S8: Undisturbed rainfall-runoff events Kruskal Wallis Test p-value results for winter, melt, and summer hydrologic conditions for the season time-varying hydrologic control. Shading indicates rejection of the null hypothesis at a significance level of 0.05. In shaded cells, an indicator marks the significantly different condition and no indicator marks all conditions being significantly different from the Dunn Test. Eta squared values as an effect size indicator are in parenthesis for significant results.**

| Rainfall-runoff event metric | Arroyo Seco | Ash Canyon Creek | Cache La Poudre River | Camp Creek | Clear Creek | Shitike Creek | Thompson River | Valley Creek | Wet Bottom Creek |
|---|---|---|---|---|---|---|---|---|---|
| volume | 0.48 | 0.88 | <0.001 (0.11) | 0.17 | <0.001 * (0.12) | 0.85 | <0.001 (0.078) | <0.001 * (0.14) | 0.008 (0.032) |
| RR | 0.32 | 0.30 | <0.001 (0.074) | 0.43 | <0.001 * (0.087) | <0.001 ^ (0.052) | <0.001 ^ (0.12) | <0.001 (0.20) | 0.06 |
| peak | 0.013 (0.017) | 0.002 ^ (0.058) | <0.001 (0.31) | 0.33 | <0.001 (0.37) | <0.001 (0.099) | <0.001 ^ (0.45) | <0.001 ^ (0.58) | 0.005 (0.037) |
| Rise | 0.05 | 0.96 | <0.001 (0.12) | 0.51 | <0.001 (0.072) | 0.33 | <0.001 ^ (0.093) | <0.001 ^ (0.17) | 0.011 (0.029) |
| Fall | 0.03 (0.012) | 0.89 | <0.001 (0.28) | 0.17 | <0.001 (0.079) | 0.28 | <0.001 (0.18) | <0.001 (0.19) | 0.10 |
| Rise_percent | 0.04 (0.009) | 0.20 | <0.001 (0.017) | 0.98 | <0.001 # (0.034) | 0.001 (0.023) | 0.82 | 0.10 | 0.14 |
| Fall_percent | 0.09 | 0.04 (0.022) | <0.001 (0.12) | 0.05 | <0.001 (0.095) | <0.001 ^ (0.031) | 0.47 | <0.001 # (0.033) | 0.04 (0.014) |
| RiseRate | 0.02 (0.013) | 0.42 | <0.001 (0.12) | 0.34 | <0.001 ^ (0.040) | 0.54 | <0.001 (0.14) | <0.001 ^ (0.16) | 0.27 |
| FallRate | <0.001 (0.037) | 0.78 | <0.001 (0.21) | 0.05 | <0.001 (0.072) | 0.02 # (0.012) | <0.001 ^ (0.13) | <0.001 (0.21) | 0.04 (0.014) |
| duration | 0.15 | 0.35 | 0.04 (0.005) | 0.61 | <0.001 # (0.064) | 0.24 | 0.16 | <0.001 # (0.042) | 0.23 |

**RC2 HESS Review Responses**

| | | | | | | | | | |
|---|---|---|---|---|---|---|---|---|---|
| rising_dur | 0.70 | 0.27 | 0.99 | 0.36 | 0.001 * (0.018) | 0.84 | 0.58 | 0.33 | <0.001 (0.065) |
| falling_dur | 0.04 (0.010) | 0.94 | <0.001 (0.029) | 0.92 | <0.001 # (0.10) | 0.002 (0.023) | 0.011 (0.035) | <0.001 (0.13) | 0.26 |
| RT1 | 0.47 | 0.04 (0.024) | 0.87 | 0.19 | <0.001 # (0.0.39) | <0.001 # (0.029) | 0.005 # (0.042) | 0.02 (0.013) | 0.28 |
| RT2 | 0.45 | 0.28 | 0.65 | 0.9 | 0.014 (0.011) | 0.008 (0.016) | 0.34 | 0.006 (0.019) | 0.36 |
| T2P1 | 0.34 | 0.14 | 0.9 | 0.17 | <0.001 # (0.031) | 0.75 | 0.95 | 0.38 | 0.001 (0.055) |
| T2P2 | 1.0 | 0.19 | 0.95 | 0.47 | <0.001 (0.023) | 0.24 | 0.17 | 0.09 | 0.006 (0.035) |
| T2P3 | 0.005 (0.023) | 0.61 | 0.58 | 0.24 | <0.001 # (0.021) | <0.001 * (0.035) | 0.003 ^ (0.049) | 0.002 (0.025) | 0.40 |
| volume_area | 0.48 | 0.88 | <0.001 (0.11) | 0.17 | <0.001 * (0.12) | 0.85 | <0.001 (0.078) | <0.001 * (0.14) | 0.008 (0.032) |
| RR_area | 0.32 | 0.30 | <0.001 (0.074) | 0.43 | <0.001 * (0.087) | <0.001 ^ (0.052) | <0.001 ^ (0.12) | <0.001 (0.20) | 0.06 |
| Peak_area | 0.013 (0.017) | 0.002 ^ (0.058) | <0.001 (0.31) | 0.33 | <0.001 (0.37) | <0.001 (0.099) | <0.001 ^ (0.45) | <0.001 ^ (0.58) | 0.005 (0.037) |
| Rise_area | 0.05 | 0.96 | <0.001 (0.12) | 0.51 | <0.001 (0.072) | 0.33 | <0.001 ^ (0.093) | <0.001 ^ (0.17) | 0.011 (0.029) |
| Fall_area | 0.02 (0.012) | 0.89 | <0.001 (0.28) | 0.17 | <0.001 (0.079) | 0.28 | <0.001 (0.18) | <0.001 (0.19) | 0.10 |
| Rise_percent_area | 0.04 (0.009) | 0.20 | <0.001 (0.017) | 0.98 | <0.001 # (0.034) | 0.001 (0.023) | 0.82 | 0.10 | 0.14 |

| | | | | | | | | | |
|---|---|---|---|---|---|---|---|---|---|
| Fall_percent_area | 0.09 | 0.04 (0.023) | <0.001 (0.12) | 0.05 | <0.001 (0.095) | <0.001 ^ (0.031) | 0.47 | <0.001 # (0.033) | 0.04 (0.014) |
| RiseRate_area | 0.05 (0.012) | 0.96 | <0.001 (0.12) | 0.51 | <0.001 (0.072) | 0.33 | <0.001 ^ (0.093) | <0.001 ^ (0.17) | 0.011 (0.029) |
| FallRate_area | 0.02 | 0.89 | <0.001 (0.28) | 0.17 | <0.001 (0.079) | 0.28 | <0.001 (0.18) | <0.001 (0.19) | 0.10 |
| duration_area | 0.15 | 0.35 | 0.04 (0.005) | 0.61 | <0.001 # (0.064) | 0.24 | 0.16 | <0.001 # (0.042) | 0.23 |
| rising_dur_area | 0.70 | 0.27 | 0.99 | 0.36 | 0.001 * (0.018) | 0.84 | 0.58 | 0.33 | <0.001 (0.065) |
| falling_dur_area | 0.04 (0.010) | 0.94 | <0.001 (0.029) | 0.92 | <0.001 # (0.10) | 0.002 (0.023) | 0.011 (0.035) | <0.001 (0.13) | 0.26 |
| RT1_area | 0.47 | 0.04 (0.024) | 0.87 | 0.19 | <0.001 # (0.039) | <0.001 # (0.029) | 0.005 # (0.042) | 0.02 (0.013) | 0.28 |
| RT2_area | 0.45 | 0.28 | 0.65 | 0.90 | 0.014 (0.011) | 0.008 (0.016) | 0.34 | 0.006 (0.019) | 0.36 |
| T2P1_area | 0.34 | 0.14 | 0.90 | 0.17 | <0.001 # (0.031) | 0.75 | 0.95 | 0.38 | 0.001 (0.055) |
| T2P2_area | 1.0 | 0.19 | 0.95 | 0.47 | <0.001 (0.023) | 0.24 | 0.17 | 0.09 | 0.006 (0.035) |
| T2P3_area | 0.005 (0.023) | 0.61 | 0.58 | 0.24 | <0.001 # (0.021) | <0.001 * (0.035) | 0.003 ^ (0.049) | 0.002 (0.025) | 0.40 |

Seasons: *Winter, ^Melt, #Summer

**Table S9: Undisturbed rainfall-runoff events Kruskal Wallis Test p-value results for none, low, and high hydrologic conditions for the antecedent precipitation time-varying hydrologic control. Shading indicates rejection of the null hypothesis at a significance level of 0.05. In shaded cells, an indicator marks the significantly different condition and no indicator marks all conditions being significantly different from the Dunn Test. Eta squared values as an effect size indicator are in parenthesis for significant results.**

| Rainfall-runoff event metric | Arroyo Seco | Ash Canyon Creek | Cache La Poudre River | Camp Creek | Clear Creek | Shitike Creek | Thompson River | Valley Creek | Wet Bottom Creek |
|---|---|---|---|---|---|---|---|---|---|
| volume | 0.55 | 0.31 | 0.98 | 0.27 | 0.34 | 0.31 | 0.09 | 0.26 | 0.09 |
| RR | 0.42 | 0.15 | 0.83 | 0.30 | 0.04 (0.016) | 0.59 | 0.13 | 0.34 | 0.31 |
| peak | <0.001 + (0.070) | 0.68 | 0.03 (0.008) | 0.71 | 0.05 | 0.26 | 0.06 | 0.20 | 0.87 |
| Rise | 0.33 | 1.00 | 0.83 | 0.78 | 0.34 | 0.17 | 0.09 | 0.51 | 0.18 |
| Fall | <0.001 + (0.055) | 1.00 | 0.02 (0.009) | 0.58 | 0.11 | 0.17 | 0.07 | 0.40 | 0.88 |
| Rise_percent | 0.63 | 0.68 | 0.14 | 0.73 | 0.26 | 0.25 | 0.09 | 0.57 | 0.07 |
| Fall_percent | 0.074 | 0.68 | 0.46 | 0.10 | 0.23 | 0.22 | 0.17 | 0.73 | 0.99 |
| RiseRate | 0.11 | 0.31 | 0.44 | 0.88 | 0.29 | 0.37 | 0.15 | 0.86 | 0.71 |
| FallRate | <0.001 + (0.078) | 0.68 | 0.04 (0.007) | 0.76 | 0.33 | 0.12 | 0.73 | 0.41 | 0.95 |
| duration | 0.29 | 0.54 | 0.33 | 0.15 | 0.15 | 0.70 | 0.17 | 0.10 | 0.11 |
| rising_dur | 0.18 | 0.54 | 0.08 | 0.39 | 0.01 + (0.033) | 0.53 | 0.44 | 0.13 | 0.04 (0.023) |
| falling_dur | 0.51 | 0.68 | 0.91 | 0.30 | 0.84 | 0.37 | 0.04 (0.18) | 0.13 | 0.61 |
| RT1 | 0.33 | 0.84 | 0.02 (0.011) | 0.85 | 0.32 | 0.19 | 0.04 (0.18) | 0.23 | 0.09 |

**RC2 HESS Review Responses**

| | | | | | | | | | |
|---|---|---|---|---|---|---|---|---|---|
| RT2 | 0.52 | 0.41 | 0.34 | 0.58 | 0.10 | 0.37 | 0.027 (0.24) | 0.97 | 0.59 |
| T2P1 | 0.08 | 0.54 | 0.03 (0.008) | 0.54 | 0.01 (0.036) | 0.93 | 0.73 | 0.09 | 0.02 (0.038) |
| T2P2 | 0.22 | 0.41 | 0.04 (0.006) | 0.71 | <0.001 (0.041) | 0.47 | 0.17 | 0.10 | 0.03 (0.029) |
| T2P3 | 0.36 | 0.22 | 0.04 (0.007) | 1.00 | 0.01 (0.040) | 0.12 | 0.13 | 0.91 | 0.46 |
| volume_area | 0.55 | 0.31 | 0.98 | 0.27 | 0.34 | 0.31 | 0.09 | 0.26 | 0.09 |
| RR_area | 0.42 | 0.15 | 0.83 | 0.30 | 0.04 (0.016) | 0.59 | 0.13 | 0.34 | 0.31 |
| Peak_area | <0.001 + (0.070) | 0.68 | 0.03 (0.008) | 0.71 | 0.05 (0.041) | 0.26 | 0.06 | 0.20 | 0.87 |
| Rise_area | 0.33 | 1.00 | 0.83 | 0.78 | 0.34 | 0.17 | 0.09 | 0.51 | 0.18 |
| Fall_area | <0.001 + (0.055) | 1.00 | 0.02 (0.009) | 0.58 | 0.11 | 0.17 | 0.07 | 0.40 | 0.88 |
| Rise_percent_area | 0.63 | 0.68 | 0.14 | 0.73 | 0.26 | 0.25 | 0.09 | 0.57 | 0.07 |
| Fall_percent_area | 0.07 | 0.68 | 0.46 | 0.10 | 0.23 | 0.22 | 0.17 | 0.73 | 0.98 |
| RiseRate_area | 0.33 | 1.00 | 0.83 | 0.78 | 0.34 | 0.17 | 0.09 | 0.51 | 0.18 |
| FallRate_area | <0.001 + (0.055) | 1.00 | 0.02 (0.009) | 0.58 | 0.11 | 0.17 | 0.07 | 0.40 | 0.88 |
| duration_area | 0.29 | 0.54 | 0.33 | 0.15 | 0.15 | 0.70 | 0.17 | 0.10 | 0.11 |
| rising_dur_area | 0.18 | 0.54 | 0.08 | 0.39 | 0.01 + (0.033) | 0.53 | 0.44 | 0.13 | 0.04 (0.023) |
| falling_dur_area | 0.51 | 0.68 | 0.91 | 0.30 | 0.84 | 0.37 | 0.04 (0.18) | 0.13 | 0.61 |
| RT1_area | 0.33 | 0.84 | 0.02 | 0.85 | 0.32 | 0.19 | 0.04 | 0.23 | 0.09 |

RC2 HESS Review Responses

| | | | | | | | | | |
|---|---|---|---|---|---|---|---|---|---|
| RT2_area | 0.52 | 0.41 | (0.011) 0.34 | 0.58 | 0.10 | 0.37 | (0.18) 0.03 (0.24) | 0.97 | 0.59 |
| T2P1_area | 0.08 | 0.54 | 0.03 (0.008) | 0.54 | 0.01 + (0.036) | 0.93 | 0.73 | 0.09 | 0.02 (0.038) |
| T2P2_area | 0.22 | 0.41 | 0.04 (0.006) | 0.71 | <0.001 + (0.041) | 0.47 | 0.17 | 0.10 | 0.03 (0.029) |
| T2P3_area | 0.36 | 0.22 | 0.04 (0.007) | 1.00 | 0.01 + (0.040) | 0.12 | 0.13 | 0.91 | 0.46 |

Antecedent precipitation: &None, ~Low, +High

7. It seems that each of the time-varying hydrologic controls (TVHC) effect on effect metrics were done individually (Table 3). I am curious how the results would look if a statistical model were created that included all of the TVHCs in it as covariates and used that to calculate effect sizes on the effect metrics.

You are correct that the analysis of each of the hydrologic controls was done individually with the aim of "identify[ing] significant time-varying hydrologic controls on rainfall-runoff event response" as defined in question 2 (lines 75-76). We were interested in this because most often in the hydrologic rainfall-runoff event and post-wildfire literature, these controls are not individually split out and the individual effects remain relatively unknown, particularly in the study area (lines 54-55, 61-67). We were also not focused on developing predictive models of runoff response, for which the recommended statistical model would be appropriate. The relative effect of the three investigated time-varying hydrologic controls and the ability to predict runoff response metrics as a function of these controls are interesting questions that future work could explore, however fall outside the scope of this study.

**Specific Comments**
**Introduction**

1. Line 33: I suggest using a reference from an academic journal.

This citation is a United States Congressional Research Report that has been cited 120 times (per Google Scholar). This citation provides context to the concern and importance of wildfire in the USA. No change has been made.

2. Line 33: "… in a changing climate the occurrence and severity of wildfire is increasing…" is this generally true? It seems like an oversimplification. Are there regions where wildfire frequency is decreasing. See "Global trends in wildfire and its impacts: perceptions versus realities in a changing world" by Doerr and Santin 2016.

You are correct that this statement is a generalization, as there are many factors and timescales that make assessing post-fire response to climate change challenging. There is a large and robust body of literature that has identified an increase in occurrence and severity in recent decades in the western US. Additionally, this recent increase has been attributed to climate change rather than fuels, see "Impact of anthropogenic climate change on wildfire across western US forests" (Abatzoglou & Williams, 2016). We have updated the text to clarify the timescale and area of interest.

Lines 33-36: "Further, with a changing climate the observed occurrence and severity of wildfire has increased in the western USA in recent decades, presenting growing challenges for human

and water security (Abatzoglou et al., 2021; Abatzoglou & Williams, 2016; Hallema et al., 2018; Murphy et al., 2018; Robinne et al., 2021)."

3. Line 33: It would be useful to include in the introduction a reference about how climate change might cause some watersheds to transition from being snow-melt fed to predominately precipitation.

Thank you for the recommendation. Based on comments by both reviewers we have edited and streamlined the introduction including an additional comment on the impacts of climate change on the precipitation regime of watersheds is outside the scope of this work.

*Study Watersheds*

4. Line 106: Unclear why this reference is here. Is this a dataset? A recommendation to use 15-minutes time series?

We have updated the text to clarify the use of the Falcone (2011) reference as the GAGES-II dataset.

Lines 81-83: "First, we identified western USA watersheds from the GAGES-II dataset (Falcone, 2011) with at least 20 years of continuous 15-minute streamflow data including at least 10 years of undisturbed streamflow including from wildfire (MTBS, 2023)."

5. Line 110: Can you include a table that details how these filtering criterion reduced the number of watersheds at each step? Started with 10k watersheds -> 5000 when filter criteria 1 applied -> 200 when filter criteria 2 was applied -> criteria 3 -> … -> criteria n.

We have updated the text identifying the selection of the study watersheds to streamline the selection process.

Lines 80-100: "Nine study watersheds in the western USA were hand-selected to satisfy a wide range of watershed properties and streamflow regimes from those with streamflow data availability (Fig. 1 a). First, we identified western USA watersheds from the GAGES-II dataset (Falcone, 2011) with at least 20 years of continuous 15-minute streamflow data including at least 10 years of undisturbed streamflow including from wildfire (MTBS, 2023). The selected nine study watersheds spanned a large range of watershed characteristics (Table 1). The contributing areas ranged over three orders of magnitude, from 14 km$^2$ (Ash Canyon Creek) to 2,966 km$^2$ (Cache La Poudre River), with extents defined by the installation locations of the long-term USGS gauges. The mean annual streamflow ranged from 12.1 m$^3$s$^{-1}$ in Thompson River to 0.03 m$^3$s$^{-1}$ in Camp Creek. The mean annual precipitation ranged from 157 cm in Shitike Creek to 53 cm in Cache La Poudre River (Falcone, 2011) and the mean annual potential evapotranspiration ranged from 780 cm in Wet Bottom Creek and 401 cm in Valley Creek (Falcone, 2011). The

watersheds included a range of streamflow regimes including seven snow melt dominated systems with average annual hydrograph peak dates between April and June and two wet season rain dominated systems with average annual hydrograph peak dates between January and February.

Two of the nine study watersheds were selected for a more in-depth exploration of watershed disturbance on rainfall-runoff events: Arroyo Seco and Clear Creek (Fig. 1 b, c). . These watersheds were selected first and foremost because they both experienced wildfires during the period of available streamflow record that burned a significant portion of the watershed (>25%) and with particularly high severity. The Station Fire (2009) burned 100% of Arroyo Seco (78% high and moderate burn severity) and the Twitchell Canyon Fire (2010) burned 25% of Clear Creek (15% high and moderate severity) (MTBS, 2023). Additionally, these two case studies provided an interesting comparison with respect to watershed characteristics, as they are an order of magnitude difference in area, are rain vs. snow-melt dominated respectively, and have a four-fold difference in mean annual streamflow."

6. Line 131: A more representative quantity (instead of the precipitation at the centroid) might be the sum/mean precipitation over the boundaries of the watershed [addressing this is of extremely high importance in my opinion].

This is something that we have thought about in depth and tested extensively. For this analysis we used the best available data and methods to quantify precipitation across the watershed. We evaluated a range of precipitation data sources, including available rain gage and different gridded precipitation datasets, and a variety of summary methods. We found that using the centroid of the watershed for the AORC precipitation dataset worked the best with our available computational power. Summarizing across the watershed took a significant amount of computation time that was not realistic for this analysis.

Future work, as identified in the discussion on lines 435-438, could expand on the precipitation quantification and summary methods used. We have expanded on this within the discussion. Additionally, we are currently developing methods to streamline the computational power required for a follow-up analysis that investigates the spatial variability of precipitation across the study watersheds.

Lines 435-438: "The centroid of the watershed was used here as the best available method given the large computational requirement for additional watershed analysis, but future work could incorporate watershed averaged precipitation or other methods to capture precipitation spatial variability (Giani et al., 2022a; Kampf et al., 2016; Wang et al., 2023)."

7. Line 135: Unclear how linear interpolation is being used here. You have hourly data and are filling in at 15 minute time steps? The interpolation algorithm should be described in crystal clear detail here.

We have updated the text to clarify the use of linear interpolation of precipitation.

Lines 116-119: "Linear interpolation was used to develop an instantaneous precipitation record at the AORC resolution of 1 mm by identifying uniform sub-timesteps within the hour timestep resolution. For example, hourly precipitation of 2 mm depth was uniformly spread over the hour with two timestamps of 1 mm each."

8. Line 138: What is the significance/relevance of 2011? And what quantity is being compared? Is this analysis justifying an assumption made from the data? It might be good to include this as supplementary material if so.

We performed this comparative analysis on the Clear Creek watershed for the water year 2011 because at the time of submission there was no published comparison of the AORC precipitation dataset in the mountainous western USA as we are using it in this study. In this watershed, 2011 was the wettest year on record and also the first-year post-fire. Since we submitted our manuscript, there has been a published comparative analysis of the AORC dataset with other relevant precipitation datasets for a similar size watershed in the western USA (Colorado, USA specifically), see "Opportunities and challenges for precipitation forcing data in post-wildfire hydrologic modeling applications" (Partridge et al., 2024). We have streamlined the text and included reference to the Partridge et al., (2024) study.

Lines 119-122: "The AORC data product was selected because of comparable or higher correlation between the AORC data product and rain gage measurements compared to other gridded precipitation data products in studies in a mountainous area in Colorado, USA, Louisiana, USA, and the Great Lakes basins (Hong et al., 2022; Kim & Villarini, 2022, Partridge et al., 2024)."

*Methods*

9. Line 153: Sorted how?

Additional clarification text included linking sorted rainfall-runoff events to the previous sentence.

Lines 130-133: "The hydrologic conditions associated with each time-varying hydrologic control were identified and assigned for each rainfall-runoff event as described in section 3.2. The assigned rainfall-runoff events were then sorted by hydrologic condition and explored as described in section 3.3."

10. Line 154: Unnecessary to bring up the use of LOWESS in this subsection."

Text was simplified per the comment.

Line 134: "Trends in rainfall-runoff event patterns were identified and inferential statistics were used …"

11. Line 157: The last sentence seems like it should be its own subsection and the selection of these two watersheds seems arbitrary.

The investigation of the two watersheds, Arroyo Seco and Clear Creek, as the two burned watersheds in this analysis has been clarified throughout the paper in response to comments by both reviewers. We have kept the sentence noted by the reviewer as is within the general introduction of the methods, but the methods applicable to only these two watersheds has been separated out into subsection 3.4 *Statistical assessment in wildfire disturbed watersheds*.

Lines 135-136: "The influence of wildfire was then evaluated relative to the undisturbed runoff event significant condition group trends in two burned watersheds, Arroyo Seco and Clear Creek."

12. Line 175: This seems sensible, except the time scale of precipitation is unclear (see comment for Line 135)

Clarification text was included in section 3.1 *RREDI toolkit*. See response to specific comments 7 and 8.

Line 155-159: "In step 1 of the RREDI toolkit, rainfall-runoff event pairs and the associated event window were identified using daily streamflow and precipitation data based on the co-occurrence of separately identified rainfall events by separating precipitation time-series into rainfall and runoff events using signal processing theory from the overlapping period of record (Fig. 2). Rainfall events were characterized by the duration, depth, and 60-minute intensity."

13. Line 182: "were also normalized" The specific methods used here are vague and are not reproducible.

Clarifying text was added to specify that normalizing was done on the basis of watershed contributing area (see revised text below). Other examples where this method has been employed and thus no additional details on the methods were included in the paper include Zheng et al. (2023), Tarasova et al. (2018), (Long & Chang, 2022), and Biederman et al. (2022).

Lines 168-170: "Metrics were also normalized by dividing metric values by the respective watershed contributing area to facilitate comparison between study watersheds."

14. Line 184: "diurnal cycling" this is never defined or discussed in the introduction.

Clarification text included to define diurnal cycling within section 3.1. *RREDI Toolkit*. This is not relevant to be included within the introduction as the comment suggested but diurnal cycling is discussed within the Discussion as a limitation of other automated rainfall-runoff event identification methods.

Lines 170-173: "Finally, in step 4, event flagging was performed to remove incorrectly identified rainfall-runoff events falling within four event identification issues: gaps in 15-minute streamflow data, diurnal cycling identified by regular daily rises and falls of flow commonly due to irrigation or snow melt cycles (Fig. S5), duplicate rainfall-runoff events, and no identified runoff event end time (Fig. 2; Fig. S3)."

15. Line 195: I don't think "systematic assessment" is an appropriate description of what is happening here. You are seemingly visually inspecting the time series for three kinds of water years (wettest/driest/typical).

Your understanding of the method is correct. Rainfall-runoff events were selected systematically in the sense that we specifically selected those in the wettest, mean, and driest years to perform the visual inspection. Included clarifying text to state this was a systematic visual inspection.

Lines 184-199: "A visual assessment of the RREDI toolkit performance was iteratively completed for all RREDI-identified rainfall-runoff events within the wettest, mean, and driest water years for each study watershed to systematically assess the RREDI toolkit performance. These years were selected based on the watershed average total precipitation from PRISM (Oregon State University, 2022). For each rainfall-runoff event, the runoff start, peak, and end timing and magnitude identified by the RREDI toolkit were visually compared with the runoff start, peak, and end timing and magnitude independently identified by visual inspection for each rainfall-runoff event following similar performance assessment methods used for other event separation methods (Giani et al., 2022b; Patterson et al., 2020; Tarasova et al., 2018b). A rainfall-runoff event was determined to be accurately identified by the RREDI toolkit if the runoff start, peak, and end magnitude and timing of each rainfall-runoff event were sufficiently similar to those timings identified through independent visual assessment such that the rise in runoff from the start to the peak and the runoff duration were considered reasonable. In this manner, we visually assessed 11% of rainfall-runoff events used in this study (774 rainfall-runoff events), that spanned a range of watersheds, watershed wetness conditions, and seasons. RREDI toolkit performance assessment results were summarized for each study watershed and across study watersheds (section 4.1). Performance results included the percent of RREDI-identified rainfall-runoff events within the wettest, mean, and driest water years with accurately identified

timing output from the RREDI toolkit, the percent of rainfall-runoff events flagged in step 4, and the percent of rainfall-runoff events retained after removal of flagged rainfall-runoff events."

16. Line 199: "assessed with respect to the four event identification issues described above" (1) I eventually figured out what was meant by "four event identification issues" but (2) how they are "assessed" is vague. You could remove this and just keep the next sentence to avoid any confusion.

We have changed as suggested.

Lines 184-199: "A visual assessment of the RREDI toolkit performance was iteratively completed for all RREDI-identified rainfall-runoff events within the wettest, mean, and driest water years for each study watershed to systematically assess the RREDI toolkit performance. These years were selected based on the watershed average total precipitation from PRISM (Oregon State University, 2022). For each rainfall-runoff event, the runoff start, peak, and end timing and magnitude identified by the RREDI toolkit were visually compared with the runoff start, peak, and end timing and magnitude independently identified by visual inspection for each rainfall-runoff event following similar performance assessment methods used for other event separation methods (Giani et al., 2022b; Patterson et al., 2020; Tarasova et al., 2018b). A rainfall-runoff event was determined to be accurately identified by the RREDI toolkit if the runoff start, peak, and end magnitude and timing of each rainfall-runoff event were sufficiently similar to those timings identified through independent visual assessment such that the rise in runoff from the start to the peak and the runoff duration were considered reasonable. In this manner, we visually assessed 11% of rainfall-runoff events used in this study (774 rainfall-runoff events), that spanned a range of watersheds, watershed wetness conditions, and seasons. RREDI toolkit performance assessment results were summarized for each study watershed and across study watersheds (section 4.1). Performance results included the percent of RREDI-identified rainfall-runoff events within the wettest, mean, and driest water years with accurately identified timing output from the RREDI toolkit, the percent of rainfall-runoff events flagged in step 4, and the percent of rainfall-runoff events retained after removal of flagged rainfall-runoff events."

17. Line 241: Effect of WYT [on?].

We have clarified the text as suggested.

Lines 245-247: "The non-parametric Mann Whitney U Test was used to evaluate the effect of WYT between the two hydrologic conditions, and the non-parametric Kruskal Wallis Test was used to evaluate the effect of season and antecedent precipitation between three hydrologic conditions, all at a 95% confidence level."

18. Line 244: Good opportunity to remind readers of how many event metrics there are and to provide an in-text citation of a list of these metrics (or explicitly list them out if there are not too many).

We have cited the table containing the list of metrics included as suggested.

Lines 249-250: "The null hypothesis for all tests was that hydrologic conditions did not impact rainfall-runoff event metrics (Table S3)."

19. Line 250: See multiple comparisons problem. The relevance of a "significance rate" is not clear either.

Please see the response to general comment 5.

20. Line 260-265: inscrutable

We have restructured the methods section to make it more clear by breaking out the wildfire statistical assessment methods (3.4 Statistical assessment in wildfire disturbed watersheds). We have added clarifying text to better describe the development of the significant condition groups and calculations of the post-fire rainfall-runoff events percentages above the undisturbed trends. We believe that this restructuring and added clarifying text addresses the comment.

Lines 266-282: "Additional statistical methods were performed on two burned study watersheds, Arroyo Seco and Clear Creek, to further explore the influence of wildfire disturbance relative to other time-varying hydrologic controls (Q3; Fig. 2). Arroyo Seco and Clear Creek were contrasting watersheds, with differing watershed characteristics, notably contributing area and streamflow regimes (Table 1) and burn characteristics (Fig. 1 b, c). For this analysis, rainfall-runoff events were defined as undisturbed or disturbed, where disturbed rainfall-runoff events were those occurring within six years post-fire (Ebel et al., 2022; Wagenbrenner et al., 2021). For the two watersheds, specific significant condition groups were identified for the rainfall depth and runoff peak relationship. To do this, the undisturbed rainfall-runoff events in each watershed were sorted into hydrologic condition permutations of the significant hydrologic controls for peak runoff. A power trend was fit to each permutation using ordinary least squares regression. The significant condition groups were identified by combining the permutations with similar power trends. An updated power trend was fit to each significant condition group. The influence of the wildfire disturbance on event runoff response was then evaluated relative to each significant condition group undisturbed trend and standard deviation. The percentage of wildfire disturbed rainfall-runoff events above the significant group trend and one standard deviation was calculated for all post-fire years combined and individually. The calculated post-fire rainfall-runoff event percents were compared to the expected 50% above the trend and 16% above the standard deviation."

*Results*

> 21. Line 267: Let the numbers and the readers decide if that's true. Delete.

Good suggestion, this change has been made.

Lines 285-286: "The RREDI toolkit resulted in a rainfall-runoff event dataset of 5042 rainfall-runoff events across the nine study watersheds (Table S4)."

> 22. Line 270: Getting lost as to what "events" are storms? Runoff? Similarly, I am confused why accuracy has anything to do with this. I was under the impression the number of events were already known and were used to extract parameters from the precipitation-stream flow time series.

We streamlined the text to refer only to 'rainfall events' for precipitation and removed 'storm events' for clarity as they had the same meaning in this manuscript as we are only considering non-snow influenced storm events. The terms "rainfall-runoff events", "runoff events", and "time-series events" are also consistently used throughout to refer to event pairs or streamflow. The rainfall-runoff event accuracy is a metric of the performance of the RREDI toolkit, an automated time-series event separation method using time-series signal processing (not a supervised learning method). The calculation of the rainfall-runoff event accuracy as a performance assessment is described in section 3.1. *RREDI toolkit*. The text has been updated to clarify the definition of accuracy as used in this paper and remind the reader of the calculation of rainfall-runoff event accuracy within the results section 4.1. *RREDI toolkit performance*. Additionally, see our response to general comments 3 and 4 regarding the methods and application of the RREDI toolkit and specific comment 25 regarding the evaluation of the RREDI toolkit.

Lines 184-199: "A visual assessment of the RREDI toolkit performance was iteratively completed for all RREDI-identified rainfall-runoff events within the wettest, mean, and driest water years for each study watershed to systematically assess the RREDI toolkit performance. These years were selected based on the watershed average total precipitation from PRISM (Oregon State University, 2022). For each rainfall-runoff event, the runoff start, peak, and end timing and magnitude identified by the RREDI toolkit were visually compared with the runoff start, peak, and end timing and magnitude independently identified by visual inspection for each rainfall-runoff event following similar performance assessment methods used for other event separation methods (Giani et al., 2022b; Patterson et al., 2020; Tarasova et al., 2018b). A rainfall-runoff event was determined to be accurately identified by the RREDI toolkit if the runoff start, peak, and end magnitude and timing of each rainfall-runoff event were sufficiently similar to those timings identified through independent visual assessment such that the rise in

runoff from the start to the peak and the runoff duration were considered reasonable. In this manner, we visually assessed 11% of rainfall-runoff events used in this study (774 rainfall-runoff events), that spanned a range of watersheds, watershed wetness conditions, and seasons. RREDI toolkit performance assessment results were summarized for each study watershed and across study watersheds (section 4.1). Performance results included the percent of RREDI-identified rainfall-runoff events within the wettest, mean, and driest water years with accurately identified timing output from the RREDI toolkit, the percent of rainfall-runoff events flagged in step 4, and the percent of rainfall-runoff events retained after removal of flagged rainfall-runoff events."

Lines 288-290: "Accuracy rates were calculated based on the comparison of the RREDI toolkit identified and independently visually identified runoff event start, peak, and end timing. Rainfall-runoff events were identified at a 69% accuracy rate pre-flagging (step 2) and the accuracy rate rose to 90% after flagging (step 4)."

23. Line 284: From my understanding of the methods, this is not a reproducible threshold identification since it is based on visual inspection. Another analyst may look at the data and conclude that the break should be different.

Please see our more extensive response to general comment 4 regarding study repeatability. The methods to reproduce the results presented in Fig. 5 are documented within the methods and Supplemental Information. We clarified the text in section 4.2 *Hydrologic variability* to indicate the method of result interpretation used in this statement.

Lines 304-306: "A slope break was visually identified at approximately 10 mm rainfall depth, above which the runoff peak increases more rapidly with increasing rainfall depth."

24. Line 295: Clear how? Was a statistical test performed? Line 298 suggests that only descriptive statistics were used to reach this conclusion.

The direction of the shifts between hydrologic conditions are described by the median as you comment. However, the significance of the difference between hydrologic conditions was tested using statistical hypothesis testing. We have removed the use of the word "clear" to leave it to reader interpretation.

Lines 316-317: "Directional shifts that varied by runoff metric and watershed were apparent in four selected runoff metric undisturbed rainfall-runoff event distributions for WYT, season, and antecedent precipitation."

*Discussion*

25. Line 398: Again, not clear how classification rates applies to this analysis as this seems to be a supervised learning problem (I think you knew whether a time period was a rain-runoff event). If that isn't the case, I am concerned that there is some circular logic going on here with the accuracy statistics (a rain-runoff event is defined as one that is identified by the RREDI algorithm, which will by definition produce an optimistic estimate of accuracy). At any rate, to provide a reliable estimate of accuracy it is important to not "double dip" and to give the algorithm "new data" to assess its performance. Common methods for this would be the hold-out validation, k-folds cross validation, monte carlo cross validation, bootstrap, etc.

The RREDI toolkit was developed to automatically separate rainfall-runoff events for any watershed based on covarying streamflow and precipitation time-series. It is not a supervised learning problem per se because we did not train the RREDI toolkit based on known rainfall-runoff events. This is because the "actual" runoff start, peak, and end timing of each rainfall-runoff event had to be determined through visual assessment and thus is not feasible to be performed for a large set of rainfall-runoff events. Rather, we applied a time-series signal processing approach as similarly applied by Giani et al. (2022b) by adjusting an existing seasonal event-detection algorithm (Patterson et al., 2020) to identify individual rainfall-runoff events. We then iteratively assessed performance at a subset of RREDI toolkit-identified rainfall-runoff events spanning different watersheds, seasons, and very dry to very wet years. What we term as "accuracy" in this paper, is the comparison of algorithm-identified runoff event start, peak, and end timings relative to those timings independently visually identified by the authors. A cross-validation is therefore not an appropriate method to apply here. We have clarified the text within the methods section 3.1 *RREDI toolkit* and included further discussion within section 5.1 *RREDI toolkit*.

Lines 184-199: "A visual assessment of the RREDI toolkit performance was iteratively completed for all RREDI-identified rainfall-runoff events within the wettest, mean, and driest water years for each study watershed to systematically assess the RREDI toolkit performance. These years were selected based on the watershed average total precipitation from PRISM (Oregon State University, 2022). For each rainfall-runoff event, the runoff start, peak, and end timing and magnitude identified by the RREDI toolkit were visually compared with the runoff start, peak, and end timing and magnitude independently identified by visual inspection for each rainfall-runoff event following similar performance assessment methods used for other event separation methods (Giani et al., 2022b; Patterson et al., 2020; Tarasova et al., 2018b). A rainfall-runoff event was determined to be accurately identified by the RREDI toolkit if the runoff start, peak, and end magnitude and timing of each rainfall-runoff event were sufficiently similar to those timings identified through independent visual assessment such that the rise in runoff from the start to the peak and the runoff duration were considered reasonable. In this manner, we visually assessed 11% of rainfall-runoff events used in this study (774 rainfall-runoff events), that spanned a range of watersheds, watershed wetness conditions, and seasons. RREDI

toolkit performance assessment results were summarized for each study watershed and across study watersheds (section 4.1). Performance results included the percent of RREDI-identified rainfall-runoff events within the wettest, mean, and driest water years with accurately identified timing output from the RREDI toolkit, the percent of rainfall-runoff events flagged in step 4, and the percent of rainfall-runoff events retained after removal of flagged rainfall-runoff events."

26. Line 403: I really think using a spatial aggregate is appropriate here because of the limitations you identify here. Please do this in the next analysis.

Please see response to specific comment 6.

---

## Author Response (AR2)

Reviewer Responses to HESS comments re:

**Leveraging a time-series event separation method to disentangle time-varying hydrologic controls on streamflow – Application to wildfire-affected catchments**

**The comments from HESS and the reviewers are in black and our responses are in blue.

**Contents**

**Referee 1**
**General Comments**

The manuscript is clearly-structured and main points are stated. The author has made efforts to revise and restructure the paper. However, there are a few issues that need to be addressed before it is ready for publication.

Thank you for taking the time to review and provide comment.

**Specific Comments**

(1)Title: For the revised paper, if you only look the title, I thought your focus of this paper is developing the event separation method and exploring the streamflow hydrologic controls. However, after reading the abstract and introduction, it feels like the overall writing style is still quite focused on exploring the impact of wildfire on streamflow which is you didn't mentioned at all in the title. If you have retained the analysis related to wildfire influences, I suggest modifying the title to something like "Leveraging a time-series event separation method to untangle time-varying hydrologic controls on streamflow—application to wildfire-affected catchments."

We have updated the title per the comment.

(2)Introduction: In current introduction, you only mention the impact of watershed disturbances

to the streamflow and the potential controls on the streamflow. Since developing this event separation method is one of the main objectives of your paper, the literature review about the current event separation methods should be added and you also need to mention why you want to develop a new event separation method? What is your motivation to do this?

We have updated the text to introduce the existing event separation methods and provide motivation for the development of the method presented in this work.

Lines 61-75: Investigating large samples of rainfall-runoff events requires automated, transferable methods for time-series event separation. Common rainfall-runoff event separation techniques rely on established baseflow methods to isolate event flow (e.g. Chapman & Maxwell, 1996; Duncan, 2019; Eckhardt, 2005; Xie et al., 2020). Runoff events are then identified where baseflow diverges from total flow (Long & Chang, 2022; Mei & Anagnostou, 2015; Merz et al., 2006; Merz & Blöschl, 2009; Tarasova et al., 2018b). Giani et al. (2022b) identified the need for increased method transferability across watersheds as the baseflow separation methods require multiple calibrated parameters in each watershed. To increase transferability, separation methods use fewer modifying watershed parameters (Blume et al., 2007; Nagy et al., 2022) or time-series signal processing to identify rainfall-runoff events (Giani et al., 2022b; Patterson et al., 2020). The commonly used separation methods are not able to identify sub-daily rainfall-runoff events as many are developed or calibrated to use only daily streamflow (Long & Chang, 2022; Mei & Anagnostou, 2015; Merz et al., 2006; Merz & Blöschl, 2009; Tarasova et al., 2018b). These methods cannot capture the sub-daily rainfall-runoff events that may result from convective rainfall events in mountainous watersheds (Kampf et al., 2016). Further, there are limitations in the existing available separation methods including the lack of identification of rainfall events with no runoff response and the filtering of diurnal cycling influenced runoff events that have limited the application of the available methods in snow-dominated watersheds.
.

(3)Line 63: 'Significant' is an adjective, and it should be followed by a noun, for example, changed to 'significant factors'.

We have updated the text per the comment.

(4)Table 1: The streamflow, P and PET values in this table look quite strange. The unit for mean annual streamflow/P/PET should be mm/year. The streamflow is listed in m²/s, which seems incorrect. Did you intend to use m³/s instead? In addition, typically, catchment PET values are between 500-2000 mm/year, and then they won't exceed 200 cm/year. Why are your PET values so large, almost an order of magnitude larger than the precipitation values, and nearly 10 times bigger? These annual data might be available in CAMELS-US dataset, it will be worth to carefully check on this.

We have updated the table 1 and associated text per the comment. The values and units for mean annual PET and mean annual precipitation have been verified against the Gages-II (Falcone, 2011) dataset. The units for mean annual PET have been corrected to mm and the values for mean annual precipitation have been converted to mm for consistency. The Gages-II dataset (Falcone, 2011) is a commonly used and accepted catchment attributes dataset for USGS gages in the USA. Precipitation and PET values from the Gages-II database were compared for three watersheds also within the CAMELS-US dataset. Differences in precipitation and PET values were attributed to differences in the estimation method used by each dataset. The values for mean annual streamflow have been converted to mm for consistency across variables. The values listed for Arroyo Seco, have been cross validated with Bart (2016).

(5)Conclusion: Can you remove the mention of Q1/Q2/Q3 in this section? I don't think it's necessary.

We have updated the text per the comment.